# Jupiter-like planets might be common in a low-density environment

**Raffaele Gratton** [1] ✉, **Dino Mesa**[1,9], **Mariangela Bonavita**[1,2,9], **Alice Zurlo**[3,4,5,9], **Sebastian Marino** [6,9], **Pierre Kervella** [7,9], **Silvano Desidera**[1,9], **Valentina D'Orazi**[1,8,9] **& Elisabetta Rigliaco**[1,9]

Radial velocity surveys suggest that the Solar System may be unusual and that Jupiter-like planets have a frequency < 20% around solar-type stars. However, they may be much more common in one of the closest associations in the solar neighbourhood. Young moving stellar groups are the best targets for direct imaging of exoplanets and four massive Jupiter-like planets have been already discovered in the nearby young β Pic Moving Group (BPMG) via high-contrast imaging, and four others were suggested via high precision astrometry by the European Space Agency's Gaia satellite. Here we analyze 30 stars in BPMG and show that 20 of them might potentially host a Jupiter-like planet as their orbits would be stable. Considering incompleteness in observations, our results suggest that Jupiter-like planets may be more common than previously found. The next Gaia data release will likely confirm our prediction.

One of the main questions in the field of extra-solar planets is how common are planetary systems similar to our own[1,2]; very different answers have been proposed so far[3,4] none of which are entirely satisfying. Our Solar System, beyond the so-called ice line (at about 3 au[5–7]) where ice particles can survive disruption by star irradiation, is dominated by giant planets. Jupiter lies at 5.2 au from the Sun. Hereinafter, we will call Jupiter-like giant planets those objects with masses $M$ larger or equal than $1M_{\text{Jupiter}}$ lying at a distance from their stars between 3 and 12 au (roughly from one to a few times that of the ice line). Note that the lower limit to the mass range is not the one defining giant planets, which is typically set around $0.3M_{\text{Jupiter}}$ to include Saturn (see Table 1). We adopt a higher lower limit because the discovery of planets with a mass $< 1M_{\text{Jupiter}}$ beyond the ice line of solar-type stars is very difficult with most of the available techniques. Models predict that giant planets should easily form around solar-type stars through the core-accretion mechanism[8–10] and that the final semi-major axis distribution should be a consequence of the position of the ice line[9],

only partially modified by migration[11]. Jupiter-like planets should therefore be common. On the other hand, most radial velocities (RV) surveys found a rather low frequency of Jupiter-like planets around solar-type stars, with occurrence rates ranging from 6% to 20% (see Table 1). We notice that these various studies use different definitions of Jupiter-like planets, as spelled in Table 1. Cuts and extrapolations are required to obtain homogeneous values. In the case e.g. of Cumming et al. study[12], integration of the best power laws over masses and semi-major axis indicates that 5.5% of the solar-type stars should have a Jupiter-like planet according to our definition. This is however an extrapolation because the upper limit of the period range considered by these authors corresponds to a semimajor axis of 3.1 au for a star of $1M_{\odot}$. There is more overlap with the sample of Wittenmyer et al.[13] that extends the semi-major axis range up to 7 au. The results of this last paper are broadly consistent with those of Cumming et al.[12], with a higher incidence rate of about 25% (well within the error bars) if the same range of masses/separation is considered. An even better overlap

¹INAF-Osservatorio Astronomico di Padova, Vicolo dell'Osservatorio 5, Padova I-35122, Italy. ²School of Physical Sciences, The Open University, Walton Hall Milton Keynes MK7 6AA, UK. ³Instituto de Estudios Astrofísicos, Facultad de Ingeniería y Ciencias, Universidad Diego Portales, Av. Ejército 441, Santiago, Chile. ⁴Escuela de Ingeniería Industrial, Facultad de Ingeniería y Ciencias, Universidad Diego Portales, Av. Ejército 441, Santiago, Chile. ⁵Millennium Nucleus on Young Exoplanets and their Moons (YEMS), Santiago, Chile. ⁶Department of Physics and Astronomy, University of Exeter, Stocker Road, Exeter EX4 4QL, UK. ⁷LESIA, Observatoire de Paris, Université PSL, CNRS, Sorbonne Université, Université Paris Cité, 5 place Jules Janssen, 92195 Meudon, France. ⁸Dipartimento di Fisica, Università di Roma Tor Vergata, via della Ricerca Scientifica 1, 00133 Roma, Italy. ⁹These authors contributed equally: Dino Mesa, Mariangela Bonavita, Alice Zurlo, Sebastian Marino, Pierre Kervella, Silvano Desidera, Valentina D'Orazi, Elisabetta Rigliaco. ✉e-mail: raffaele.gratton@inaf.it

**Table 1 | Frequency of Jupiter-like planets from radial velocities and microlensing surveys**

| Authors | $M$ ($M_{Jupiter}$) | $a$ (au) | N. stars | Fraction (%) | Remarks |
|---|---|---|---|---|---|
| Cumming et al.[12] | 0.3–10 | <3 | 585 | 10.5 | |
| Mayor et al.[124] | 0.3–10 | <5 | 825 | 14 | |
| Wittenmyer et al.[13] | 0.3–10 | 3–7 | 202 | $6.2^{+2.8}_{-1.6}$ | |
| Fernandes et al.[15] | 0.1–20 | 0.1–100 | | $26.6^{+7.5}_{-5.4}$ | |
| | 1–20 | 0.1–100 | | $6.2^{+1.5}_{-1.2}$ | |
| Zhu[14] | 0.3–10 | <10 | 393 | $19.2 \pm 2.8$ | |
| Bryan et al.[17] | 1–20 | 5–20 | 123 | $52 \pm 5$ | Stars with inner planets |
| Gould et al.[22] | | | | $16 \pm 16$ | Microlensing |

First column gives the references; the second column is the planet mass range considered (in units of the Jupiter Mass); $a$ is the semi-major axis range (in au) and N. stars is the number of stars used to derive the fraction, that is given in Column 5. Column 6 gives remarks.

is with Zhu[14] which extends the semi-major axis range to 10 au. This last study gives a higher incidence of giant planets than expected using the best value for the exponents in the Cumming et al.[12] power laws. The excess is by a factor of two if we consider the total population of planets and 1.6 if we consider the stars that have at least one planet. This corresponds rather well to the incidence of $6.9 \pm 1.8\%$ ($2.2 \pm 0.8\%$) of planets with mass in the range $1$–$10 M_{Jupiter}$ ($3$–$10 M_{Jupiter}$) and semi-major axis $3$–$10$ au from Fig. 4 of Zhu paper[14]. A frequency slightly lower than this is obtained by Fernandes et al.[15] who considered both RV and Kepler planets. If we integrate their asymmetric, mass-dependent frequency over our definition of Jupiter-like planets, we obtain a frequency of 4.2%; it is only 1.7% if we consider the range $3$–$10 M_{Jupiter}$ and semi-major axis $3$–$10$ au. This last analysis should, however, be taken with some caveat, because it assumes that mass distribution is independent of period, a result which might not be valid[16]. We might summarise that the frequency of planets in the mass range $3$–$10 M_{Jupiter}$ and semi-major axis $3$–$10$ au obtained from RV is about $2.0 \pm 1.0\%$.

A higher total occurrence rate of planetary companions detected through RV of $52 \pm 5\%$ was obtained by Bryan et al.[17] in their sample of stars having an inner planet. However, while very interesting, this higher incidence may be not considered an estimate of the overall frequency of Jupiter-like companions because the conditional probability of a particular star hosting a cold Jupiter, given the existence of a hot/warm Jupiter, is likely not random[14,17–21]. This shows how inner and outer planets are not independent as their formation conditions and system evolution are connected. Finally, from a single microlensing event of a giant planet at wide separation, Gould et al.[22] derived a frequency of about 16%. These results were interpreted as evidence that systems similar to the Solar System are not common (about 15% and likely <20%)[1,2,4]. It should be noticed here that the frequency of giant planets depends on stellar mass[23] and metallicity[24]. We will return to these points in the "Discussion" section.

The RV survey results have an important statistical weight; however, the underlying population is likely to be a wide mix of objects formed in different environments, in most cases several Gyr ago. These birth environments are largely unknown. In addition, the systems might have undergone a significant evolution since their formation. This makes a comparison between these results and theoretical prediction rather difficult. The observation of Jupiter-like planets in very young associations represents a possible solution to these issues. In this context, the BPMG[25] is an ideal target for high-contrast imaging (HCI) because it is the closest, and one of the youngest (age about 20 Myr[26,27]) associations of A-F stars in the solar neighbourhood. Its proximity and young age, make it the region where the direct detection of Jupiter-like planets is by far easiest. Most of the directly imaged sub-stellar companions detected in the BPMG in fact belong to this category, with a resulting detection yield much higher than what is typically obtained for HCI surveys[28,29]. However, selection effects are

still very large and the real frequency is likely to be much higher than the measured detection rate. This suggests that Jupiter-like planets can be very common in the BPMG, a result that seems odd with the RV results and needs to be properly settled and discussed.

In this work, we examine in more detail the evidence obtained for the BPMG.

## Results

### Frequency of substellar companions

Shkolnik et al.[25] give a total of 146 members to the BPMG. We focused here on the objects with masses $M > 0.8 M_\odot$ (mass range from 0.80 to $2.2 M_\odot$, with a median value of $1.15 M_\odot$). The actual list of members of the BPMG considered in this paper is discussed in the "Methods" section, subsection "Sample selection". The final sample is made of 30 stars. It can be noticed that the membership of some of these stars in the BPMG is uncertain. We adopted a conservative approach, where some uncertain members around which no companion was detected were nonetheless kept in the final list; while this reduces the derived frequency of Jupiter-like planets in the BPMG, the effect is <10%. While there are low mass members even closer to the Sun (such as AU Mic), the parallaxes of the stars more massive than the $0.8 M_\odot$ in the BPMG range from 51.44 to 12.00 mas (distances between 19.4 and 83.3 pc), with a median value of 21.47 mas (distance of 46.6 pc). At this median distance, the maximum apparent separation from the star of a planet in circular orbit with a semi-major axis similar to that of Jupiter (5.2 au) would be 112 mas. This is very close to the outer edge of the coronagraph for state-of-the-art high-contrast imagers such as the Spectro-Polarimetric High-contrast Exoplanet REsearch (SPHERE)[30] and the Gemini Planet Imager (GPI)[31]. This means not only that detection of a Jupiter-like planet is very difficult in an individual observation, but also that it can be behind the coronagraphic mask for a large part of or even the whole orbit. Repeated observations are then needed to increase the chance of observing it at the right phase for detection (see e.g. the case of AF Lep b[32–34]). The implication is that the search of Jupiter-like planets using HCI is likely severely incomplete as HCI surveys typically do not re-observe a star in case of a lack of detection of candidate companions.

To put the detection of several Jupiter-like planets in the BPMG into a context, we constructed a list of all known companions to the stars with $M > 0.8 M_\odot$, either stellar or substellar. Details are given in the "Methods" section, subsection "Stellar and substellar companions". We considered 30 entries, but actually, in three cases separate entries are given for the components of wide binaries where both stars have a mass $>0.8 M_\odot$, implying that we are considering a total of 27 systems. If we only consider stellar companions, we found that 12 of these systems are binaries and 5 are triples, and the remaining 10 are single. The fraction of single stars ($37 \pm 12\%$) is lower than that found for solar-type stars by Raghavan et al.[35], as expected because this last study refers to much older stars and we expect some fraction of binaries to dissipate

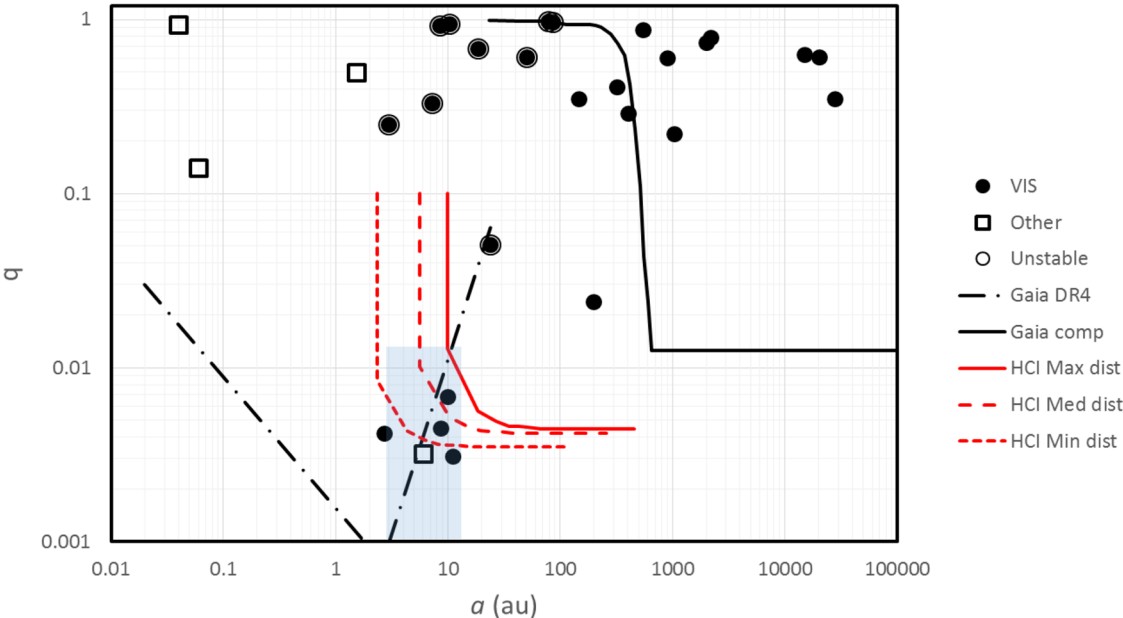

**Fig. 1 | Mass ratio $q = M_B/M_A$ as a function of the semi-major axis $a$ (in au) for companions (both stellar and substellar) to stars in the BPMG.** Filled circles are objects detected in imaging; open squares are companions detected using other techniques. Companions circled are those that make unstable the orbits of Jupiter-like planets. The solid, long-dashed and short-dashed red lines represent the detection limits in typical high-contrast imaging observations for the stars farthest, median, and closest to the Sun, respectively. The solid black line marks the detection limit of Gaia[130] as separate entries; the dash-dotted black line is the predicted astrometric detection limit of Gaia DR4 release (expected not before the end of 2025) for G-dwarfs at 50 pc[81,82], comparable to the median distance of the BPMG members. The light blue area is the one corresponding to the Jupiter-like planets. Source data are provided as a Source Data file.

as they age. It is similar to that of other low-density young associations (CrA[36]; Taurus-Auriga[37,38]).

With this data, we may construct the plot of separation vs. mass ratio $q = M_B/M_A$ between the components shown in Fig. 1; $M_A$ and $M_B$ are the masses of the primary and secondary, respectively. Similar plots are widely used to discuss planet formation scenarios[39]. In the case of the BPMG, the plot shows a few interesting features. There are only 8 out of 23 stellar companions that are M-type stars (here $M < 0.5 M_\odot$). This is a fraction of $35 \pm 10\%$. For comparison, objects with mass $M < 0.5 M_\odot$ are in the range of 62–77% for the most commonly considered stellar initial mass functions[40–42]. Considering that the search is complete down to much lower masses, this is quite a low fraction, suggesting a correlation between the mass of the primary and that of the secondary, as found e.g. for stars in the Sco-Cen association[43]. All the substellar companions are in a restricted region around the star; in particular, the five planetary companions are between 3 and 12 au, and have a narrow range of mass ratios $0.003 < q < 0.008$ (this is close to the detection threshold, and may then reflect selection effects). They are then all Jupiter-like planets and this strongly suggests they formed in a very similar way, very likely by core-accretion[8,9,44,45]. Note that all these planets would be undetectable at a distance >100 pc such as that of e.g. the Sco-Cen and Taurus associations, which are both much richer and younger than the BPMG, because they would appear projected too close to their stars. Hence, the short distance to the BPMG is crucial for their discovery. The two brown dwarfs (BDs) PZ Tel B and $\eta$ Tel B are at rather larger separations (tens to hundreds au) and may belong to a separate group with a different formation mechanism (most likely disk instability[46,47]). Similar objects would be easy to detect in Sco-Cen and Taurus using HCI.

We can immediately notice that the observed incidence of Jupiter-like planets around the BPMG stars considered in this paper is much higher than expected from RV surveys. In fact, if we limit to the mass range to 3–10$M_{Jupiter}$, Zhu[14] gives a frequency of only $2.2^{+0.9}_{-0.7}\%$ of the stars having companions in the range of 3–10 au (admittedly, slightly smaller than the range 3–12 au that we considered for Jupiter-like

planets). This makes very unlikely (probability of $5 \times 10^{-4}$) the detection of 5 companions over a sample of 30 stars as found in the BPMG. The discrepancy would be even larger if we use the frequency of Jupiter-like planets obtained using data from Fernandes et al.[15]. However, in the following discussion we intend to show that the discrepancy is even larger than this.

To better estimate the implications of our result on Jupiter-like planets, we should consider in fact a few additional facts. A fraction of the stars in the BPMG have stellar or BD companions at a separation likely incompatible with the stability of the orbit of a Jupiter-like planet. To this purpose, we applied the stability criteria by Holman et al.[48] to all the systems including a stellar or BD companion. This formula requires knowledge of the orbital eccentricity $e$, which is known only for PZ Tel[49]. For the remaining targets, we assumed an average value appropriate for the apparent separation of the targets given by the relation:

$$<e> = -0.0117 \log a^3 + 0.0529 \log a^2 + 0.07 \log a + 0.3226, \quad (1)$$

where $a$ is the semi-major axis in au. This was obtained by combining the eccentricity distribution for close binaries obtained by Murphy et al.[50] with that for wide binaries by Hwang et al.[51]. Strictly speaking, the relation between the average projected separation and the semi-major axis that we used to estimate this last quantity for several targets depends on the assumed eccentricity[52], so an iterative procedure should be required here. However, corrections to the value of the eccentricity that is obtained by considering this fact are always < 0.005, and the variation of the semi-major axis is always < 5% for our targets. We will then neglect these second-order effects. With this assumption, we found that the orbits of Jupiter-like planets cannot be stable in ten systems (TYC 1208-468-1, HIP 76629, TYC 6820-0223-1, HD 161460, HIP 88726, CD-64 1208, HD 173167, HIP 92680 = PZ Tel, TYC 6878-195-1, HIP 103311). The case of HIP 98839 is quite marginal because the orbits of Jupiter-like planets would be unstable if the eccentricity is larger than $e = 0.67$. However, we will assume here conservatively that Jupiter-like planets may exist in this system. This

leaves a sample of 20 stars that might potentially host a Jupiter-like planet including the very close binary HIP 84586 = HD 155555 for which such planets would be in a stable circumbinary configuration.

### Orbital sampling effect

Planets at this short separation from the stars are hidden by the coronographic mask for a large fraction of their orbit and when this happens they cannot be detected using HCI. We will call this the orbital sampling effect. We used two independent approaches to address this point. The first one is an estimate of the expected completeness of existing HCI data. The second one uses the astrometric signal obtained comparing data from the Hipparcos[53] and Gaia[54] satellites to provide an indication of the presence of a Jupiter-like planet. The two approaches give a very similar correction.

**Completeness of HCI detections.** To estimate the relevance of the orbital sampling effect, we will assume that the semi-major axis of the planets in this sample of 20 stars distributes uniformly over the range 3–12 au and that only companions seen at separation larger than 150 mas at the epoch of the observation could be detected (this is the minimum separation where there was a detection of a planetary companion around the BPMG stars). We then estimated for each star the probability that a planet is observed at separation >150 mas running a Monte Carlo code with 100,000 random extractions in phase along the orbit. We considered two eccentricity distributions (circular and uniform over the range 0–0.5, this last reproducing the known distribution for Jupiter-like planets, see the "Methods" section, subsection "Eccentricity distribution for Jupiter-like planets") and random inclination $i$. We took into account the number of epochs at which each target was observed either by SPHERE or GPI. This was done by counting the observations with SPHERE listed in the ESO Archive and the GPI observations listed by Nielsen et al.[28]. We assumed here that the observations were acquired at a one-year separation (note that the orbital periods typically are a few tens of yr). The fraction of planets detected at least once—that is, the probability of detecting a Jupiter-like planet—is given in Table 1 (we only give the result for the eccentric orbit distribution). It depends on the distance from the Sun and on the number of HCI observations. It ranges from values close to 1 for $\beta$ Pic (that was observed many times and is close to the Sun) to about 0.037 for HIP 89829, which is the object farthest from the Sun and has a single HCI observation, and 0 for HIP 107620 that was never observed with modern HCI instruments. The mean probability that a planet in the range of separation 3–12 au around one of the 20 stars possibly hosting Jupiter-like planets is not behind the coronagraph in at least one of the epochs when it was observed is 40.4% for circular orbits and 42.3% for orbits with eccentricities uniformly distributed between 0 and 0.5. We used this last value hereinafter.

So far four Jupiter-like planets have been discovered around three systems in the BPMG from HCI campaigns; they are all more massive than $4 M_{\text{Jupiter}}$. Applying the orbital sampling correction, this implies that $36 \pm 19\%$ of the stars with mass $M > 0.8 M_\odot$ and that potentially have Jupiter-like planets have one (or more) detected planet with a mass $>4 M_{\text{Jupiter}}$ and with semi-major axis in the range 3–12 au.

**Indirect indications through astrometry signal.** An additional indication for the presence of companions is given by the Proper Motion Anomaly (PMa)[55,56]; this can be used for those systems where planets were not detected in HCI. The PMa is the difference in the Proper Motion measured in the Gaia Data Release 3 (DR3) catalogue and the secular motion obtained comparing Gaia and Hipparcos positions. Additional estimates of the PMa were obtained considering the Hipparcos and Gaia Data Release 2 proper motions, but they are less accurate than that obtained using Gaia DR3 values[57,58]. On the other hand, also the Gaia re-normalised unit weight error (RUWE) can be used to constrain the characteristics of potential companions. The

RUWE parameter is a measure of the residuals around the best five-parameter astrometric solution[59]. A low value of the RUWE indicates that the companion responsible for the PMa is not massive and/or is close to the star[60]. We can then derive the upper limits of the possible masses of companions at short separation using the procedure described in the "Methods" section, subsection "Upper mass limits from the Gaia RUWE parameter".

The relevant data for this discussion are given in Table 2. Entries are the 20 stars where there is no massive companion that would make the orbit of Jupiter-like planets unstable[48]. Note however that only 17 of these stars were included in the Hipparcos catalogue; for the remaining three stars there is not any information about the PMa. For each star, we give the signal-to-noise ratio SNR of the PMa and the value of the companion mass indicated by Kervella et al.[55] (this is the average of the values obtained at 5 and 10 au), the number of HCI observations available, the probability that the companion is at separation >150 mas in at least one such observation, and for those objects that were actually detected through HCI, the semi-major axis of the orbit and masses derived from the orbits and from their luminosity (using the 20 Myr isochrones by Baraffe et al.[61,62]). In the case of the eclipsing binary HIP 84586 = HD 155555, we revised the companion mass derived from the PMa taking into consideration that the star is a close binary and summing the masses of the two components.

If we consider as significant a PMa detected with an SNR(PMa) > 3 by Kervella et al.[55], the following additional targets are likely to have planetary-like companions: HIP 560, HIP 10679, HIP 84586, HIP 88399 (see (see section "Methods", subsection "Stars with significant Proper Motion anomaly"). These four stars have all been observed in HCI without detection of close companions; however, the lesson of HIP 25486 = AF Lep b, detected by Mesa et al.[32] after previous unsuccessful attempts, tells that the planets may well be there, but they are not detectable in some observations because at that epoch they were projected too close to the star. Indeed, for these targets, the probability that the planet is observed when it is not hidden by the coronagraph is rather low (see Table 2). This probability decreases with the distance from the Sun. Figure 2 shows the distribution of the stars of the BPMG with $M > 0.8 M_\odot$ in the distance mass plane. We used different symbols for stars around which a Jupiter-like planet has been detected using HCI, there is an indication of a similar companion from PMa, there is a binary companion that makes unstable the orbits of potential Jupiter-like planets, there is no detection, and for those lacking PMa data. Jupiter-like planets have been discovered around the three closest stars of the BPMG with mass above solar. Those stars where there is an indication for similar (yet undetected) companions from PMa are intermediate in distance and mass. This figure shows that the planets discovered so far in the BPMG are around the three A-F stars closest to the Sun. It is then very reasonable that there are Jupiter-like planets around the four stars with significant PMa not yet detected in HCI because they are seen projected too close to the star - or too faint to be detected. Indeed, four further indirect detections coincide with the correction to the actual number of detected planets expected due to orbital sampling. The median mass provided by Kervella et al.[55] for them is $5.2 M_{\text{Jupiter}}$, with individual values in the range 3.28–18.45 $M_{\text{Jupiter}}$. They are difficult objects for HCI, but still, they are planets much more massive than Jupiter.

If we add these four indirect detections to those obtained with HCI, the frequency of stars hosting Jupiter-like planets with masses $>4 M_{\text{Jupiter}}$ is $41 \pm 12\%$ (7 out of 17), if we only consider those stars that have PMa values around the stars that might possibly host them. The frequency of Jupiter-like planets may even be larger because there may be similar or smaller planets that do not cause a significant PMa, as suggested by the cases of $\beta$ Pic and of 51 Eri that have a low SNR(PMa) though they are known to host planets.

**Table 2 | Data for stars of the BPMG possibly hosting a Jupiter-like planet**

| HIP | $\pi$ (mas) | SNR (PMa) | M (PMa) ($M_{Jupiter}$) | $N_{obs}$ | Prob | a (au) | $M_{dyn}$ ($M_\odot$) | $M_{ev}$ ($M_\odot$) | Remark |
|---|---|---|---|---|---|---|---|---|---|
| 560 | 25.16 | 3.02 | 3.28 | 2 | 0.482 | | | | |
| 10679 | 25.23 | 5.21 | 18.45 | 1 | 0.484 | | | | |
| 10680 | 25.28 | 2.04 | 4.56 | 2 | 0.487 | | | | |
| AG Tri | 24.42 | | | 1 | 0.459 | | | | |
| 21547 | 33.44 | 2.08 | 5.16 | 10 | 0.712 | 11.1 | 5.5 ± 2.6 | 3.5 ± 1.1 | 113 |
| HD286264 | 18.83 | | | 1 | 0.273 | | | | |
| 25486 | 37.25 | 8.99 | 3.91 | 3 | 0.740 | 8.6 | 5.2 ± 0.1 | 3.6 ± 1.8 | 32 |
| 27321 | 51.44 | 0.86 | 3.41 | 20 | 0.983 | 2.68 | 8.9 ± 0.8 | 7.1 ± 2.0 | 116 |
| | | | | | | 9.93 | 11.9 ± 3.0 | 11.2 ± 1.0 | 66 |
| 29964 | 25.57 | 1.52 | 1.23 | 2 | 0.496 | | | | |
| 84586 | 32.95 | 4.87 | 6.3 | 2 | 0.662 | | | | Mass of A = 2.23$M_\odot$ |
| TYC8728-2262-1 | 14.79 | | | 2 | 0.115 | | | | Mass of A = 1.12$M_\odot$ |
| 86598 | 15.20 | 2.80 | 4.77 | 2 | 0.133 | | | | |
| 88399 | 20.29 | 3.98 | 4.09 | 7 | 0.338 | 6 | 4 | | 103 |
| 89829 | 12.43 | 0.45 | 0.98 | 3 | 0.037 | | | | |
| 92024 | 32.95 | 2.46 | 7.36 | 2 | 0.669 | | | | |
| 95261 | 20.60 | 2.16 | 17.15 | 5 | 0.354 | 197.1 | | 47.6 ± 4.0 | 118 |
| 95270 | 20.93 | 2.33 | 2.45 | 3 | 0.354 | | | | |
| 98839 | 21.11 | 5.10 | 4.94 | 3 | 0.355 | | | | M-companion at 146 au |
| 99273 | 19.96 | 2.65 | 2.55 | 3 | 0.321 | | | | |
| 107620 | 30.08 | 2.34 | 1.41 | 0 | 0.000 | | | | |

Hipparcos number (HIP), the parallax $\pi$, the signal-to-noise ratio SNR(PMa), and the mass (M) obtained from analysis of the PMa[55], the number of HCI observations ($N_{obs}$), the probability (Prob) that the object is not behind the coronagraph in at least one of these observations, the semimajor axis (a) and dynamical and evolutionary mass of companions detected in HCI ($M_{dyn}$ and $M_{ev}$ respectively), and remarks appropriate to the objects.

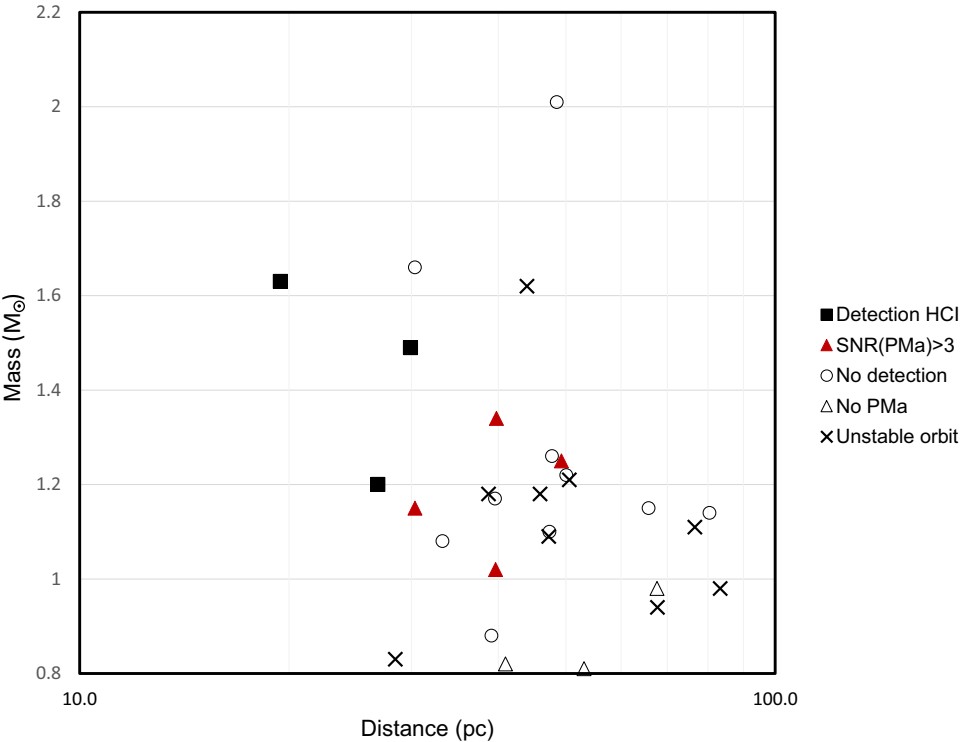

**Fig. 2 | Run of the mass of the target stars (in $M_\odot$) as a function of distance d in parsec.** Different symbols are used for those stars around which a Jupiter-like planet has been detected using HCI (black filled squares), there is an indication of a similar companion from PMa (red triangles), there is a binary companion that makes unstable the orbits of potential Jupiter-like planets (crosses), there is no detection (open circles) and those lacking PMa data (open triangles). Source data are provided as a Source Data file.

**Summary about the correction due to orbital sampling.** The two estimates of the frequency of stars having planets with masses $>4M_{Jupiter}$ and separation in the range 3–12 au around stars with mass $M > 0.8M_\odot$ are then $36 \pm 19\%$ if we consider the orbital sampling correction to HCI observations and $41 \pm 12\%$ if we combine direct detections through HCI and indirect detections through PMA. The two values agree with each other within the error bars. In the following, we will adopt an average of the two values: $39 \pm 12\%$, where the error bar is the minimum between the two different estimates. This is more than an order of magnitude larger than the value obtained from RV surveys: we in fact recall that the frequency of planets in the mass range 3–10$M_{Jupiter}$ and semi-major axis 3–10 au obtained from RV is about $2.0 \pm 1.0\%$.

## Mass correction

The total number of Jupiter-like planets in the BPMG is expected to be larger than derived in the previous analysis because a large fraction of the giant planets is expected to have a mass $<4M_{Jupiter}$ and would then be undetected using either high-contrast imaging or PMa. For instance, a real Jupiter analogue (mass ratio $q$ about 0.001, semi-major axis of 5.2 au) would still be undetectable with the current techniques even in this close group of stars. We notice that the final Gaia data release DR4 expected not before the end of 2025 will be sensitive to planets even below this mass at the distance of the BPMG stars. To estimate the relevance of the correction for planets with mass in the range 1–4$M_{Jupiter}$ we should consider the mass distribution $F(M_p)$ of the mass $M_p$ of giant planets that is however not well known. Observations[63] suggest a power-law distribution of the form of $dF/dM_p = kM_p^{-p}$, with $k$ a constant and the index $p$ is about $1.3 \pm 0.2$, over the relevant planetary mass range (and for stellar masses in the range 0.5–2$M_\odot$). We may then consider a similar distribution over the range 1–15$M_{Jupiter}$, with the upper mass end selected in order to include $\beta$ Pic b and not too much inconsistent with the range considered by Adams et al.[63], and the lower limit consistent with our definition of Jupiter-like planets. We found that $37 \pm 7\%$ of these planets should have $> 4M_{Jupiter}$. The mass correction is then a factor of 2–3. The correction for incompleteness in mass would be even higher if a steeper slope or a lower mass limit were adopted for giant planet mass function, as suggested e.g. by Nielsen et al.[28].

## Statistical relevance of the observed frequency of Jupiter-like planets in our sample

In order to assess the statistical meaning of the frequency of the occurrence of at least one Jupiter-like planet around a star in the BPMG, we did the following. We considered sequences of 20 stars and assumed a probability $x$ that each of these stars has a Jupiter-like planet. We then assumed that each of these planets has a probability $P$ to be observed, having a mass above the threshold for detection (4$M_{Jupiter}$). $P$ is extracted from a Gaussian distribution centred on 0.39 and a standard deviation equal to 0.12, to take into account the uncertainties present in the mass distribution. For each sequence, a given number of planets will then be detectable, depending on the value of $x$ and $P$. Since we have 7 actual direct or indirect detections over 20 stars (a conservative value because not all data are available for all stars), we repeated the extraction 100,000 times for each value of $x$ at the step of 0.01 and counted which fraction of the cases yielded at least 7 detections. We found that for $x = 0.99$, we expect at least 7 detections in 50% of the cases and that this happened for only 5% of the extractions when $x = 0.50$. We conclude that the best value of $x$ is $x = 0.99$ and it is $>0.50$ at a 95% level of confidence.

## Discussion

Our analysis suggests that the formation of Jupiter-like planets around stars with masses $>0.8 M_\odot$, at least for environments such as the BPMG, is common (the best estimate is 99% and it is $>50\%$ at a 95% level of confidence). This is indeed a prediction of the core-accretion models[8,9,44,45].

The frequency of Jupiter-like planets found for the BPMG is much higher than typical for FGK stars in the solar neighbourhood, which we reported in the Introduction. The metallicity of the BPMG is discussed in the "Methods" section, subsection "Properties of the BPMG". It is likely solar and then similar to the median of the stars observed in the RV surveys, so we may exclude that it plays an important role. We may think of four possible reasons for this apparent discrepancy, either individually or most likely in combination.

First, the stars that we consider in the BPMG ($M > 0.8M_\odot$, median mass 1.15$M_\odot$) have a mass that typically is larger than that considered in the RV surveys. The masses of the three stars hosting Jupiter-like planets discovered by HCI are in the range 1.2–1.63$M_\odot$; the range is more extended if we consider also stars with an SNR(PMa) $> 3$, but still, the median mass is 1.20$M_\odot$. The RV surveys provide results for stars with spectral type later than F5 and down to K9, that is in the 0.60–1.33$M_\odot$ range using data by Pecaut and Mamajek[64]. The median mass of the stars we considered in the BPMG is then roughly 20% higher than for typical field RV surveys. On the other hand, the frequency of giant planets is expected to have a linear dependence on the stellar mass from 0.4 to 3$M_\odot$[9,65], a prediction that is verified by the observations[23]. This might then explain part (though likely not all) of the observed difference.

Second, with an age of about 20 Myr, the BPMG is much younger than the stars surveyed with RVs, whose typical age is a few Gyr. The lower frequency of planets seen around old stars might reflect a progressive loss of planets from systems, either due to long-term instabilities of the systems or to the encounters with nearby objects (or both, if close encounters trigger dynamical instabilities). It is difficult to assess the relevance of this effect for the programme stars because we know little of their planetary systems. The two planets of $\beta$ Pic are quite strongly interacting with each other, though the configuration is likely stable[66].

Third, the disruptive dynamical influence of outer companions is treated in a different way in RV surveys and in the present discussion. While RV surveys have typically biases against close binaries, a significant fraction of such objects are included in the samples. As an example, 13% of the stars in the uniform detectability sample by Fischer et al.[24] (which largely overlaps with the Cumming et al.[12] one) have companions which do not allow dynamically stable orbits for planets with $a \le 10$ au[67]. Such targets would have been excluded in our statistical evaluation, somewhat reducing the observed discrepancy.

Finally, the total mass in stars of the BPMG should be about 94 M$_\odot$ (see "Methods" section, subsection "Stellar and substellar companions"). The highest mass star is $\eta$ Tel A with a mass of 2.0$M_\odot$; so it should have been a T-association at birth, though at the low end of the mass distribution[68]. Also, at odds with other nearby young associations, the BPMG does not appear to be either comoving or possibly linked to the core of known nearby open clusters[69], further supporting its low-density nature. We define here as a low-density environment a star-forming region with less than 300$M_\odot$. Miller and Scalo[70] estimated that about 20% of the stars should form in T-associations, about 60% in more massive OB-associations (most massive stars with a mass $>10M_\odot$), 10% in R-associations (most massive stars in the range 3–10$M_\odot$), and the remaining 10% in open clusters (see also Lamers et al.[71]). On a similar tone, Lada & Lada[72] spell that the vast majority (90%) of stars that form in embedded clusters form in rich clusters of 100 or more members with masses in excess of 50$M_\odot$, and embedded clusters account for a significant (70–90%) fraction of all stars formed in giant molecular clouds. So most stars likely form in more massive environments than the BPMG. It is possible and even likely that this influences the formation of Jupiter-like planets because the lifetime of disks is limited by encounters (around more massive stars) and photoevaporation (around low-mass stars, $M < 0.5M_\odot$) by nearby massive

stars[68,73–77]. This lifetime might then become comparable to or even shorter than required for the formation of giant planets in the core-accretion scenario. Adams et al.[73] proposed that encounters with potentially disruptive solar systems are frequent for very young clusters with an initial population larger than 1000 stars, while there should be virtually no effect if the population is as low as 100 stars. Winter et al.[75] found a canonical threshold for the local stellar density ($n \geq 10^4$ pc$^{-3}$, that is however much larger than typical of associations[68]) for which encounters can play a significant role in shaping the distribution of protoplanetary disk radii over a time-scale about 3 Myr. According to the planet formation simulations by Mordasini et al.[9] there should be virtually no giant planet around solar mass stars if the disk lifetime is shorter than 1.5 Myr, while they may be present around 30–40% of the stars for lifetimes of 5 Myr. This is likely connected with a result from Pawellek et al.[78], where they found that the incidence rate of debris disks around F-type stars in the BPMG is about 75% and much higher than for older field stars, which cannot be explained by simple collisional evolution models. This further adds to the possibility of a more efficient planet (and planetesimal) formation in the BPMG.

Concerning this last hypothesis, there is an active debate about the birthplace of the Solar System. It has been proposed that only two very distinct types of stellar groups are serious contestants as the cradle of the Sun—high-mass, extended associations ($M > 20{,}000 M_\odot$) and intermediate-mass, compact clusters ($M > 3000 M_\odot$)[79,80]. This is because of the need for a nearby SN to explain the isotopic abundances for pre-solar grains (e.g. the abundances of $^{26}$Al and $^{60}$Fe) and of a not-too-disrupting environment to explain its current characteristics. Possible present-day counterparts would be the association NGC 2244 and the M44 cluster. Both these possible environments have a much higher mass and density than the BPMG.

We finally notice that the huge harvest of planetary systems expected for the final release of Gaia should give a definitive answer to the frequency of Jupiter-like planets in a variety of environments in a few years from now[81,82].

## Methods
### Properties of the BPMG
There exists a significant debate within the literature regarding the precise elemental abundances of young clusters, associations, and star-forming regions. Indeed, recent inquiries have cast doubt upon the apparent absence of young stellar systems that possess solar or super-solar compositions, suggesting that this may be attributed to the influential presence of intense magnetic fields that impact the formation of spectral lines within the upper photosphere. For a comprehensive examination of this subject matter, we refer interested readers to the work of Baratella et al.[83]. Numerous high-resolution investigations indicate that young associations demonstrate a metallicity level that is approximately similar to that of the Sun, denoted as [Fe/H] = 0 in a logarithmic scale, with a conservative measurement error estimated at approximately ± 0.2 dex. Regarding the specific case of the BPMG, only one study has been conducted thus far[84], and it solely focuses on a single star within this system. Their findings reveal a value of [Fe/H] = −0.01 ± 0.08, which lends support to the notion of a young stellar population conforming to the solar composition. It should be noted that an exhaustive investigation of the metallicity of the BPMG will be expounded upon in a dedicated publication.

### Sample selection
We constructed a list of all known companions to stars in the BPMG more massive than the $0.8 M_\odot$. The stars were initially selected from the list by Shkolnik et al.[25] complemented with three further targets ($\alpha$ Cir, HIP 111170 and HIP 107620) proposed by Gagne and coworkers[85,86]. While the original list included 38 stars, we culled seven stars (HIP 7576, HIP 11360, HIP 47110, $\alpha$ Cir, HIP 79881, HIP 105441, and HIP 111170) that are not likely members according to more recent

analysis[87–89] or for which the properties of a late-type companion are clearly not compatible with the young age of BPMG ($\alpha$ Cir[90]). We also checked the membership of these targets using the BANYAN $\Sigma$ tool[88] with Gaia DR3[91] kinematic parameters and found that HIP 12545 has a null membership probability, and consequently, we did not consider it. Two stars with ambiguous membership indicators (HIP 107620 and HIP 98839) were kept conservatively in the target list (removing them will further increase the retrieved planet and disk frequencies). We did not consider V4046 Sgr which still hosts a gas-rich disk, because the planet formation phase cannot be considered as complete. $\eta$ Tel is not included in the list of members by Gagne et al.[85] and Couture et al.[27]. This is possibly due to the low-accuracy RV of the fast-rotating early type star but the properties of the very wide comoving object HIP 95270 (classified as a bona fide member in all the studies) and of the BD companion $\eta$ Tel B are fully compatible with membership and we then kept the object in the sample. The final list included 30 stars.

### Stellar and substellar companions
Companions to these stars were drawn from the following data sets:
- Visual companions from Gaia DR3 catalogue[92] (Gaia in Table 3). Gaia provides separate entries for objects with separation larger than about 0.7 arcsec, and it is sensitive down to the substellar regime at separation larger than a few arcsec. We considered companion objects with full (5-elements) astrometric solution within 10 arcmin from each star, whose relative projected velocity is below the escape velocity from the star at the projected separation. For these systems, we assumed that the semi-major axis (in au) is equal to the projected separation divided by the parallax, which corresponds to the thermal eccentricity distribution[93] of $f(e) = 2e$, $e$ being the orbital eccentricity[52]. This assumption underestimates the semi-major axis by about 25% in the case of circular orbits. We deem the uncertainties related to this fact as negligible in the present context.
- Additional visual binaries were considered from Mendez et al.[94] and from the Binary Star Catalogue[95] (VIS in Table 3).
- For objects with magnitude $G > 4$ (that is, are not heavily saturated on the Gaia detectors), the RUWE parameter[59] that describes the residuals around the 5-elements solution can be used as an indication that the star is an unresolved binary[60,96]. Objects with RUWE > 1.4 are likely binaries. However, none of the stars in our list have RUWE > 1.4, save $\beta$ Pic itself, which is however brighter than the limit where this method can be used, and the known binaries HIP 76629 = V343 Nor, CD-64 1208 and TYC 6878-195-1.
- Objects with significant Proper Motion Anomaly (PMa) from Kervella et al.[55], that we assume here to have a signal-to-noise ratio SNR(PMa) > 3 (they are marked with PMa in Table 3). The PMa is the difference between the "instantaneous" proper motion measured by Gaia (DR3, epoch 2016.0), and the "long term" proper motion that is derived by considering the Hipparcos and Gaia positions at their respective epochs (1991.25 and 2016.0). The PMa is not available for nine stars because they were not included in the Hipparcos catalogue. Ten out of the remaining 21 stars with mass > $0.8 M_\odot$ in the BPMG have a significant PMa. This fraction is not unusual for nearby stars such as those in the BPMG. PMa is sensitive well down to substellar and even planetary masses for objects as close as those in the BPMG.
- Optical interferometry from Evans et al.[97] and Absil et al.[98]. However, while these studies revealed warm debris disks around several of the BPMG, they did not detect any companion. Direct detections of giant planets have however been reported using the Very Large Telescope Interferometer VLTI/GRAVITY instrument[99,100], opening a promising parameter space.
- High-contrast imaging using either GPI at the Gemini Telescope[31] or SPHERE at VLT[30] (HCI in Table 3). High contrast imaging with

**Table 3 | Summary of data about companions to stars with $M > 0.8M_\odot$ in the BPMG**

| HIP | Other | $\pi$ (mas) | $G_A$ (mag) | $G_B$ (mag) | $K_A$ (mag) | $K_B$ (mag) | $M_A$ (M$_\odot$) | $M_B$ (M$_\odot$) | $q$ | $a$ (au) | Method |
|---|---|---|---|---|---|---|---|---|---|---|---|
| 560 | | 25.16 | 6.094 | | 5.240 | | 1.34 | | | | |
| | TYC1208-468-1 | 18.95 | 10.032 | 10.181 | | | 0.79 | 0.77 | 0.97 | 86.26 | Gaia |
| 10679 | | 25.23 | 7.623 | | 6.262 | | 1.02 | | | | |
| 10680 | | 25.28 | 6.904 | 7.623 | 5.787 | 6.262 | 1.17 | 1.02 | 0.87 | 544 | Gaia |
| | AG Tri | 24.42 | 9.744 | 11.433685 | 7.080 | | 0.82 | 0.49 | 0.60 | 900.86 | Gaia |
| 21547 | 51 Eri | 33.44 | 5.141 | | 4.537 | 18.490 | 1.49 | 0.0046 | 0.0031 | 11.1 | HCI |
| | | | 5.141 | 9.80 | 4.537 | 6.413 | 1.49 | 1.11 | 0.74 | 2002 | Gaia |
| HD286264 | V1841 Ori | 18.83 | 10.362 | | 7.637 | | 0.81 | | | | |
| 25486 | AF Lep | 37.25 | 6.210 | | 4.926 | 16.646 | 1.20 | 0.0054 | 0.0045 | 8.60 | HCI |
| 27321 | $\beta$ Pic | 51.44 | 3.823 | | 3.480 | 12.470 | 1.63 | 0.0068 | 0.0042 | 2.68 | HCI |
| | | | 3.823 | | 3.480 | 14.300 | 1.64 | 0.0112 | 0.0068 | 9.93 | HCI |
| 29964 | AO Men | 25.57 | 9.273 | | 6.814 | | 0.88 | | | | |
| 76629 | V343 Nor | 25.83 | 7.677 | | 5.852 | | 1.18 | 0.29 | 0.25 | 2.94 | VIS |
| | | | | 13.208 | | | 1.47 | 0.43 | 0.29 | 399.40 | Gaia |
| | TYC6820-0223-1 | 12.00 | 10.092 | | 8.054 | 8.226 | 0.98 | 0.93 | 0.95 | 10.33 | HCI |
| 84586 | V824 Ara | 32.95 | 6.461 | | 4.702 | | 1.15 | 1.08 | 0.94 | 0.04 | EB |
| | | | 6.46 | 11.47 | 4.702 | 7.629 | 2.23 | 0.49 | 0.22 | 1035 | Gaia |
| | TYC8728-2262-1 | 14.79 | 9.324 | | 7.364 | | 0.98 | 0.14 | 0.14 | 0.06 | SB |
| 86598 | | 15.20 | 8.195 | | 6.992 | | 1.15 | | | | |
| | HD 161460 | 13.05 | 8.864 | | 6.776 | | 1.11 | 1.03 | 0.93 | 8.43 | VIS |
| 88399 | | 20.29 | 6.912 | | 5.913 | | 1.25 | 0.0040 | 0.0032 | 6.0 | PMa |
| | | | 6.912 | 12.313 | 5.913 | 8.273 | 1.25 | 0.51 | 0.41 | 320 | Gaia |
| 88726 | HR6749 | 22.74 | 5.615 | 5.69 | 5.140 | 5.140 | 1.62 | 1.59 | 0.98 | 78.2 | Gaia |
| 89829 | | 12.43 | 8.671 | | 7.053 | | 1.14 | | | | |
| 92024 | HR7012 | 32.95 | 4.725 | 8.887 | 4.298 | 6.096 | 1.66 | 1.31 | 0.79 | 2165 | Gaia |
| | CD-64 1208 | 35.16 | 8.877 | 11.877 | 6.096 | | 0.83 | 0.28 | 0.33 | 7.16 | VIS |
| | HD173167 | 19.77 | 7.175 | | 6.136 | | 1.21 | 0.60 | 0.50 | 1.53 | SB |
| | | | 7.175 | 11.13 | 6.136 | 7.854 | 1.81 | 0.63 | 0.35 | 27812 | Gaia |
| 92680 | PZ Tel | 21.16 | 8.102 | | 6.366 | 11.716 | 1.09 | 0.056 | 0.051 | 23.6 | HCI |
| | TYC6878-195-1 | 14.77 | 9.877 | | 7.750 | 8.675 | 0.94 | 0.64 | 0.68 | 18.62 | HCI |
| 95261 | $\eta$ Tel | 20.60 | 5.012 | 6.936 | 5.010 | 11.600 | 2.01 | 0.048 | 0.024 | 197 | HCI |
| | | | 5.012 | 6.94 | 5.010 | 5.910 | 2.06 | 1.26 | 0.61 | 20210 | Gaia |
| 95270 | | 20.93 | 6.936 | | 5.910 | | 1.26 | | | | |
| 98839 | | 21.11 | 8.023 | 13.691 | 6.627 | | 1.10 | 0.38 | 0.35 | 146.01 | Gaia |
| 99273 | | 19.96 | 7.076 | | 6.076 | | 1.22 | | | | |
| 103311 | | 21.78 | 7.131 | | 5.811 | | 1.18 | 0.72 | 0.61 | 50.5 | VIS |
| | | | 7.131 | 9.91 | 5.811 | 7.039 | 1.90 | 0.73 | 0.63 | 14918 | Gaia |
| | | | 7.131 | 12.74 | 5.811 | | 1.18 | 0.46 | | | Gaia |
| 107620 | | 30.08 | 7.431 | | 6.050 | | 1.08 | | | | |

Columns are as follows: Hipparcos number, alternative star designation, parallax $\pi$ (in mas), Gaia $G$ magnitude of primary and secondary ($G_A$ and $G_B$, respectively), Two-micron All Sky Survey 2MASS $K$ magnitude of primary and secondary ($K_A$ and $K_B$, respectively), mass of primary and secondary ($M_A$ and $M_B$, respectively), mass ratio $q$, orbital semi-major axis $a$ (in au), and method used to detect the companion.

these high-performing instruments is available for all but six of the target stars, the missing objects being TYC 1208-468-1, HD 161460, HIP 98839, HD 173167, CD-26 13904, HIP 107620, some of them being known visual binaries from observations with lower contrast[101]. Data used are from various sources[28,102–106]. In the future, important contributions are also expected from James Webb Space Telescope (JWST[107]).

- Radial velocity variations (SB in Table 3). The relevant data were from Trifonov et al.[108] and Grandjean et al.[109]. We found no data in the Amber catalogue based on the Fibre-fed Extended Range Optical Spectrograph (FEROS) data[110]. We also examined the RVs measured by Gaia[111] that are available for 23 stars. All these stars have variable RVs, but in most cases, the RV variations can be explained by pulsation and/or activity.
- HIP 84586 = V824 Ara = HD 155555 is a renowned eclipsing binary (EB in Table 3). The adopted solution is from Tokovinin[95]. We also inspected the time series in the Transiting Exoplanet Survey Satellite (TESS) archive that is available for 15 of the stars but we did not find evidence of additional companions.

The parameters for the substellar companions were obtained as follows: 51 Eri b[112,113]; AF Lep b[32]; $\beta$ Pic b and c[66,114–116]; HIP 88399 b[103]; PZ Tel B[29,117]; $\eta$ Tel B[118].

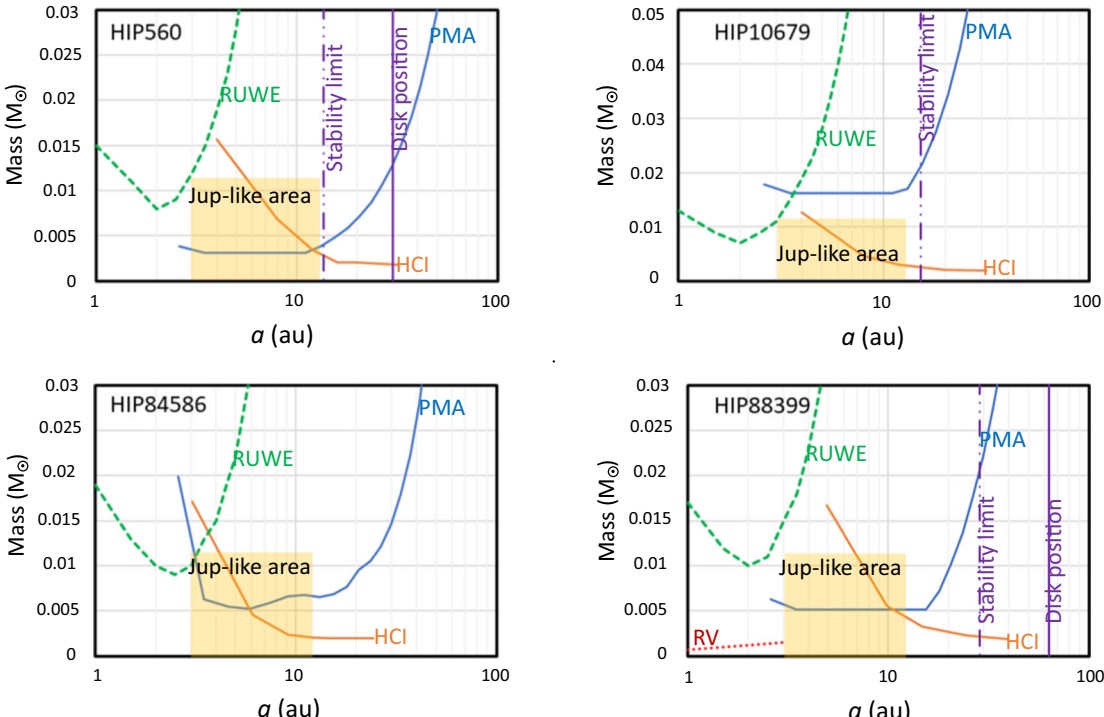

**Fig. 3 | Diagrams showing the mass of companions (in $M_\odot$) that may be responsible for the observed Proper Motion anomaly (PMA)[55] as a function of semi-major axis $a$ in au (blue solid line).** They are compared with 90% confidence upper limits obtained from the Gaia RUWE parameter (green dashed line), the upper limit from high-contrast imaging (HCI) with SPHERE[30] (orange solid line), and the upper limit from RVs (red dotted line)[109]. The solid violet lines mark the position of known debris disks[78] and the dash-dotted violet lines are the outer edge of the stability region due to the presence of these disks or of known companion (HIP 10679). Upper left panel: HIP 560, HCI from Dahlqvist et al.[106]; Upper right panel: HIP 10679, HCI from Dahlqvist et al.[106]; Lower left panel: HIP 84586 HCI from Asensio-Torres et al.[104]; Lower right panel: HIP 88399, HCI from Mesa et al.[103]. The orange area is occupied by Jupiter-like planets. The companions responsible for the PMa should be close to the solid blue line, below the dashed green, the solid orange line, the dotted red line, and the left of the dash-dotted violet line.

Masses for the stars are derived from the absolute Gaia $G$ magnitudes using the calibration by Pecaut and Mamajek[64] if $G < 3$, else from the 20 Myr old isochrone by Baraffe et al.[62]. The masses in Column 8 are the sum of the masses of all components within the considered one.

For HIP 21547, the far companion is itself a binary (GJ3305); the mass is the sum of the two components.

The secondary of HIP 88399 is at PA = 89.12 degree which is quite different from the PA of the PMa (131 ± 14 degree); the mass of $0.51 M_\odot$ is also not enough to produce a PMa as large as observed by a factor of about four. We then conclude that the PMa is caused by a closer companion, not yet detected in imaging.

In the case of HIP 98839, the PMa of the primary (both in absolute value and velocity) may be explained by the secondary detected by Gaia if its mass were about $0.43 M_\odot$, to be compared with the value of about $0.35 M_\odot$ derived from the photometry: this is a good agreement given the approximations made. The secondary has a high value of the RUWE parameter and the radial velocity reported by Gaia is offset by 25.3 km s$^{-1}$ with respect to the primary (though they have very similar parallax and proper motion, clearly indicating the physical link): these two facts indicate that it is likely itself a close binary; this would explain a dynamical mass larger than that derived from photometry. However, since we have no further details on it, we will not make any correction to its total mass for this. Anyhow, for the Occam razor, we will assume that this is the object responsible for the PMa.

The companion of HIP 92024 = HD 172555, CD-64 1208, is itself a binary[119] with an apparent separation of 0.1–0.2 arcsec from the online version of the Washington Double Star (WDS) catalogue[120]; the value for the mass ratio $q$ is that obtained by dividing the mass of

the primary for the sum of the two components of the secondary from Bonavita et al.[121].

For HIP 103311 we took into consideration that the wide Gaia companion is itself a close binary; the mass ratio $q$ with respect to the inner binary is the value obtained by combining the two components.

In addition, HIP 10679/HIP 10680, HIP 95261/HIP 95270, and HIP 92024/CD-64 1208 are wide binaries. They are given as primaries and as secondaries to the brighter component.

The total mass in these systems is 52 $M_\odot$. Summing up all the other members in Shkolnik et al.[25], the total mass of the BPMG should be about 94 $M_\odot$.

### Stars with significant Proper Motion anomaly

The following four targets are likely to have planetary-like companions: HIP 560, HIP 10679, HIP 84586, HIP 88399 because of the value of the PMa[55]. Figure 3 illustrates the constraints we can derive for these objects by combining data from the PMa, from RUWE, and HCI. The case of HIP 88399 has been described in detail by Mesa et al.[103] which included in their analysis also the information from RVs[109]. We may consider that in this system there is very likely a planet at a distance of about 6 au and a mass of about $4 M_{\text{Jupiter}}$, with uncertainties of a factor of two or less in both quantities. In the case of HIP 10679, the PMa is too large to be due to the secondary; it is also directed towards a PA (138 ± 11 degree) that is very different from that of the secondary (HIP 10680: PA = 31.81 degree), as it should rather be expected at this wide separation[122]. We then attribute it to additional companion(s). The location of the companions is further constrained by the presence of debris disks around HIP 560 and HIP 88399, and of the known companions for HIP 84586 and (HIP 10679) for HIP 10680. These set inner and outer limits to the region for stable orbits for the systems. We

**Table 4 | Eccentricities for Jupiter-like and other planets at separation < 80 au**

| Planet | M M_Jupiter | a au | e | Reference |
|---|---|---|---|---|
| Jupiter-like planets (a < 12 au) | | | | |
| beta Pic c | 6.8 | 2.68 | 0.24 | 66 |
| HD206893 c | 12.7 | 9.6 | 0.41 | 125 |
| AF Lep b | 5.4 | 8.6 | 0.47 | 32 |
| beta Pic b | 11.2 | 9.93 | 0.10 | 66 |
| 51 Eri b | 4.6 | 11.1 | 0.45 | 126 |
| Slightly farther planets (12 < a < 80 au) | | | | |
| HR8799 e | 8.1 | 16.28 | 0.148 | 127 |
| PDS70 b | 7.0 | 22.7 | 0.17 | 128 |
| HR8799 d | 9.5 | 26.78 | 0.115 | 127 |
| PDS70 c | 4.4 | 30.2 | 0.037 | 128 |
| HR8799 c | 7.7 | 41.40 | 0.054 | 127 |
| HD95086 b | 2.6 | 62 | <0.18 | 129 |
| HR8799 b | 6.0 | 71.95 | 0.017 | 127 |

Columns are as follows: Planet designation, Mass, Semimajor axis, eccentricity, references.

computed these limits as in Holman and Wiegart[48], with null eccentricity for the disks and an eccentricity equal to 0.66 for the case of a stellar companion. With these constraints, the companions of HIP 560, HIP 84586, and HIP 88399 are likely Jupiter-like planets (the area corresponding to this class of planets is marked as orange in the plots). If a single object, the companion of HIP 10679 is a small BD of about $20 M_{Jupiter}$. However, such a massive object would likely have been detected in HCI and/or through the value of RUWE. Alternatively, the large PMa value might be explained by the combination of the contributions by two massive Jupiter-like planets; this explanation may better match the lack of detection in HCI and through RUWE. The observed PMa for a star having several companions is the vector sum of the PMa due to individual companions. The signal due to two planets, each one with a mass smaller than that indicated by the blue line, may combine to produce a signal similar to an individual planet with a mass as indicated by the blue line if the individual PMa vectors have a similar direction. Note that a combination of the signal of two planets may also lead to a cancellation, that is a PMa lower than expected for the individual objects, if the individual PMa vectors are directed at very different angles one from the other.

## Debris disks

As our adopted member list has some differences with respect to Pawellek et al.[78], we update the census of frequency of F-type stars with debris disks accordingly. With the removal of HIP 11360 and HIP 111170, it results that 8/10 stars have clear signatures of the presence of debris disks (spatially resolved disks or IR excess), confirming the very high disk frequency[78]. Considering instead our full list of stars more massive than $1 M_\odot$, the frequency of stars with disks results of 60% (12/20), and, for the stars without perturbers in the 3–12 au region considered for statistical analysis for planetary companions, a higher frequency of 69% (11/16). All these frequencies do not include observational incompleteness and are then lower limits.

## Eccentricity distribution for Jupiter-like planets

There are few Jupiter-like planets discovered so far from HCI; a few more are at separation <80 au. The eccentricities of their orbits are given in Table 4. This sample is small and possibly biased; eccentricities may be overestimated when errors are large; finally, the role of multiple-planet systems (that typically have low eccentricities) is not well clear. With these caveats, we note that there is no known planet detected through

direct imaging with $a < 80$ au with $e > 0.5$ and that the average value is $e = 0.20 \pm 0.16$. This value agrees within the errors with the median eccentricity obtained for 126 Jupiter-like planets extracted from RV surveys ($e = 0.26^{+0.09}_{-0.06}$; using data listed in the Extrasolar Encyclopaedia, that are however typically much older than those considered here and may have their eccentricities pumped up by secular evolution. The low typical value for the eccentricity of planets agrees with the conclusion of Bowler et al.[123]. We will then assume that Jupiter-like planets have a uniform distribution of eccentricities over the range of 0–0.5.

## Upper mass limits from the Gaia RUWE parameter

The Gaia RUWE parameter is a measure of the residuals around the 5-parameter astrometric solution by Gaia. A value of RUWE > 1.4 is indicative of binarity[60] and corresponds to residuals $\sigma$ around the solution larger than the value of about 0.3 mas which is the typical error of individual Gaia measures[59]. We may use this to set upper limits to the mass of companions in the case where RUWE < 1.4. To this purpose, we computed the residuals around a best-fit straight line fitting through astrometric points of a sequence simulating the Gaia observations but including the wobble of the primary due to the orbital motion of a companion of different mass. For small mass objects, we may neglect the contribution due to the light from the secondary to the motion of the photocenter. For simplicity, we assumed circular orbits, but we assumed random phases (at the start of the Gaia observations) and inclinations $i$, respectively with uniform and proportional to cos $i$ distributions. For each mass of the secondary, we repeated the experiment with 1000 random trials and derived the secondary mass where 90% of the trials have $\sigma < 0.3$. We notice that the astrometric signal is given by $(M_B/M_A)s$, where $M_A$ and $M_B$ are the masses of the primary and secondary (in units of $M_\odot$), respectively, and $s$ is the semi-major axis $a$ in au multiplied by the parallax $\pi$ (in mas). We found that the 90% confidence upper limit of the mass of the secondary obtained through the Monte Carlo simulation is well reproduced by the approximate relations:

$$M_B < \sigma \frac{M_A}{a\,\pi} \quad (2)$$

if the time covered by the observations considered by Gaia DR3 $\Delta T = 2.83$ year is shorter than 0.6 times the orbital period $P$ (in yr), and:

$$M_B < \sigma \frac{M_A}{a\,\pi} \left(\frac{P}{\sqrt{2}\Delta T}\right)^2 \quad (3)$$

if $\Delta T > 0.6P$.

## Data availability

Datasets generated during and/or analysed during the current study are provided with this paper in the source_file.xls file. To generate this data we used the ESO Archive at http://archive.eso.org/eso_archive_main.html; the Gaia Data Release 3 (DR3, https://www.cosmos.esa.int/web/gaia/dr3); the TESS database https://www.nasa.gov/tess-transiting-exoplanet-survey-satellite; the Extrasolar Encyclopedia (http://exoplanet.eu/ Source data are provided with this paper.

## Code availability

We wrote two simple procedures using IDL: out_of_coro.pro: this code estimates the probability of observing a Jupiter-like planet projected out of the coronagraphic field mask using a Monte Carlo approach; best_fraction.pro: this code estimates the most probable value of the fraction of stars hosting Jupiter-like planets given the observed number of detections in the BPMG sample. Both codes are available from the first author upon request. We used the on-line BANYAN Σ tool (https://www.exoplanetes.umontreal.ca/banyan/banyansigma.php) to check the membership of the targets to the BPMG association.

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

## Acknowledgements

R.G., S.D., V.D., D.M., E.R. acknowledge support from the PRIN-INAF 2019 "Planetary systems at young ages (PLATEA)" and ASI-INAF agreement no. 2018-16-HH.O. A.Z. acknowledges support from the FONDECYT Iniciación en investigación project number 11190837 and ANID— Millennium Science Initiative Programme—Center Code NCN2021_080. This work has made use of the SPHERE Data Center, jointly operated by OSUG/IPAG (Grenoble), PYTHEAS/LAM/CeSAM (Marseille), OCA/Lagrange (Nice), and Observatoire de Paris/LESIA (Paris). This research has made use of the SIMBAD database, operated at CDS, Strasbourg,

France. This work has made use of data from the European Space Agency (ESA) mission *Gaia* (https://www.cosmos.esa.int/gaia), processed by the *Gaia* Data Processing and Analysis Consortium (DPAC, https://www.cosmos.esa.int/web/gaia/dpac/consortium). Funding for the DPAC has been provided by national institutions, in particular, the institutions participating in the *Gaia* Multilateral Agreement. This paper includes data collected by the TESS mission and obtained from the MAST data archive at the Space Telescope Science Institute (STScI). Funding for the TESS mission is provided by the NASA Science Mission Directorate. STScI is operated by the Association of Universities for Research in Astronomy, Inc., under NASA contract NAS 5-26555. This publication makes use of data products from the Two Micron All Sky Survey, which is a joint project of the University of Massachusetts and the Infra-red Processing and Analysis Center/California Institute of Technology, funded by the National Aeronautics and Space Administration and the National Science Foundation.

## Author contributions

R.G. conceived the work and prepared the manuscript, F.K. and S.M. developed the PMa concept, D.M., M.B., A.Z., V.D., and E.R. worked on the observation and analysis, S.D. worked on the definition of the star sample and their characterisation, All authors reviewed the manuscript.

## Competing interests

The authors declare no competing interests.
