## [Peer Review File · Nature Communications]

REVIEWER COMMENTS

Reviewer #1 (Remarks to the Author):

Review of Nature Comms paper:

"Jupiter-like planets are common in a low density environment" by Gratton et al. (2023)

The authors have reviewed measurements of the occurrence rates of Jupiter-like planets from various literature sources, which together suggest that Jupiter-like planets are rare, occurring around less than 20% of stars. The authors have also considered the occurrence rates of Jupiter-like planets (mass > 0.1 M_{Jup} orbiting 3.0 to 12.0 au from their star) in a nearby stellar association: the Beta Pic Moving Group (BPMG), using existing studies and data from the Gaia spacecraft. Together, those data suggest that Jupiter-like planets in the BPMG are very common (occurrence rate almost 100%). Hence the authors conclude that Jupiter-like planets are common in low-density environments such as BPMG.

The overall methods of the paper are satisfactory, the results are statistically robust, and the conclusion appears to be supported. I do have comments that will perhaps affect the quoted occurrence rate but that will not change the conclusion.

The work is important because it contrasts with existing studies with only partially overlapping parameter spaces. There is also ample relevant discussion interpreting the new results and comparing them to the literature. It deserves to be published in Nature Comms.

Detailed comments follow.

1. The studies quoted in Table 1 have used different working definitions of Jupiter-like planets to the one used in this work. All studies use a narrower mass range (typically beginning at 1.0 M_{Jup}) and a narrower orbital radius than this work. Given the broader mass range in this work, it naturally follows that the occurrence rate found here will be higher. It is also known and expected that the occurrence rate of planets is a function of semi-major axis (or orbital radius); the definition in Cumming et al. is entirely non-overlapping with this work, making a direct comparison difficult, and for the other sources it naturally follows that the authors should find more companions in their broader semi-major axis range than are found in those literature studies. I would like to see a more thorough discussion and calculation of both of the above effects on the occurrence rate comparison.

2. Regarding the Bryan et al. study, I agree with the authors that the conditional probability of a particular star hosting a cold Jupiter given the existence of other planets interior to that cold Jupiter's orbit is probably not random. However, I would like to see the evidence or a reference.

3. Regarding the mass ratio distribution of companions to stars in the 1.0 to 2.5 Msun mass range, the authors appear surprised that the distribution is skewed towards companions in the $q \sim 0.2$ to 1.0 range. In fact, this distribution has been well measured and discussed (e.g. Moe & Di Stefano 2017 for a range of primary masses, Murphy et al. 2018 for ~ 1.5 to 2.0-Msun primaries). The latter study also shows the 'thermal distribution' of orbital eccentricities (attributed to Ambartsumian 1935, cited by the authors) to be a poor representation of the observations, hence the average eccentricity of 0.67 is a poor choice to use in the authors' simulations of dynamical stability. The 'average', taken as the eccentricity where the cumulative distribution reaches 0.5, is measured by Murphy et al to be closer to 0.33. Further, a power law $p(e) \sim e^{-\eta}$ with $\eta = -0.1$ is a better fit than using a simple average, and a two-parameter Gaussian distribution with $\mu = 0.36$ and $\sigma = 0.27$ would be better still.

4. For clarity and readability, I would suggest the following. After the sentence "The median probability that a planet in the range of separation 3-12 au around one of 16 stars possibly hosting Jupiter-like planets is not behind the coronagraph in at least one of the epochs when it was observed is 39%." the authors might want to add: "that is, the probability of detecting a planet at that separation around one of the 16 stars, given the sampling, is 39%." If that is not the intended meaning, then I fear the original wording is not very clear.

5. In figure 2, the solid "red" line actually appears orange.

6. In the final paragraph before the mass correction, the authors calculate the frequency of stars hosting Jupiter-like planets to be $44 \pm 18\%$, based on 7 out of 16 stars having them. The uncertainty here might not have been calculated correctly. The stars here represent Bernoulli trials: each draw represents a discrete probability distribution, which has a value of 1 ('star has planet') with probability p , and a value of 0 ('star has no planet') with probability $q = 1 - p$. The uncertainty is then $\sqrt{p \cdot q / n} = \sqrt{0.44 \cdot (1 - 0.44) / 16} = 0.124$. Hence the frequency is $(44 \pm 12)\%$.

Following corrections to the above (the comparisons in point 1, the simulations in point 3, and the uncertainty in point 6), I anticipate that the conclusion will remain the same but the significance may differ. I look forward to seeing the update.

Reviewer #2 (Remarks to the Author):

The paper presents the fraction of solar-type stars hosting a "Jupiter-like" planet in the Beta Pictoris young moving group and concludes that the fraction is higher than found in the field.

My biggest concern is that this statistic is found by combining both confirmed *and undetected* Jupiter-like exoplanets within a small, biased sample of stars. In particular, that the number of candidate exoplanets doubles the sample of confirmed ones (4 confirmed plus 4 candidates), artificially enhancing the principal statistic supporting the claim, and ultimately that the claim is based on small number statistics.

Major comments:

- The term "low density environment" should be clearly defined early in the paper, as it could be interpreted in myriad ways depending on the subfield of expertise of the reader.
- Figure 1 is a bit cryptic. It is not clear what is the message that it is trying to convey with the orange rectangle. It would be more digestible if the rest of the exoplanet population was shown to provide context. At minimum, a color legend would be immensely helpful.
- Assuming a point estimate of $e=0.67$ for the eccentricity of Jupiter-like planets is not well motivated and unlikely to be robust. A better reference would be Bowler et al. 2020 from which the authors could draw a distribution of young, directly-imaged planets, for a close observational match to the parameters of their sample. Alternatively, a Monte Carlo simulation would serve a similar purpose to achieve robustness.
- The assumption of eccentricities for Jupiter-like planets is inconsistent across the paper: a circular orbit assumption is used to estimate the fraction of time a planet spends behind a coronagraph, and is therefore undetectable, yet a fixed eccentricity of $e=0.67$ was assumed to calculate orbital stability.
- Since the claim of high Jupiter-like frequency around solar-type stars relies on the detectability fraction estimated through their simulation, a much more detailed description and discussion of said simulation is warranted, as well as clear parameter boundaries for its applicability (e.g., range of primary and companion masses where the simulation and observations are sensitive).
- Does the "95% level of confidence" refer to a confidence interval? If yes, how was this defined? Is it a percentile of the bootstrapped samples? Do 95% of the Monte Carlo runs yield a frequency of Jupiter-like planets between 58 and 100%?
- The direct comparison between direct imaging and RV samples in the Discussion needs to be taken with a grain of salt. The BPMG is a young association, whereas the RV-detected giant planet sample is primarily composed of field-age stars since young stars have high levels of RV jitter and have been selected out of RV surveys for decades. This is an important bias to take into account (in addition to the difference in median primary mass as noted by the authors) since dynamical evolution can dramatically affect the orbital parameters of a given system and it is more likely that field-age systems have undergone chaotic dynamical evolution than systems in a young moving group.

- The discussion involving the "conundrum with the low frequency found by radial velocity surveys" needs significant development.

Minor comments:

- Please reference Franson et al. 2023 as well for the discovery of AF Lep b.

- The down selection of BPMG sample from 164 to the final 14 does not justify why they only pick stars more massive than the Sun if their goal is to explore "Sun-like" stars, especially since their analysis suffers from small number statistics.

- Figure 1 brings into question what the authors may mean with the word "companion". This word is typically used for stellar or brown dwarf companions. If used for "planetary companion," especially since they use the term to refer to objects within the orange rectangle, it would be helpful for the reader that the authors specify this explicitly.

- Says: "We considered companions objects with full (5-elements) astrometric solution within 10 arcmin from each stars, whose relative projected velocity is below the escape velocity". Should say "of the cluster" at the end of this sentence.

- This statement: "Objects with RUWE > 1.4 are likely binaries" needs a reference. I believe Torres & Stassun 2018 should do.

- The paper would be strengthened with Figures exploring the sample in more detail, for example showing the distributions of mass and distance.

- Please justify the need for Figure 1 (why is mass ratio vs semi-major axis an interesting space to explore?).

- "The stellar companions are confined in a quite narrow range of mass ratios

($0.2 < q < 1.0$)" -- Since mass ratio spans from 0-1, $q = 0.2-1.0$ is most of the mass ratio range.

- Multiple typos were found throughout the text.

Reviewer #3 (Remarks to the Author):

Review of "Jupiter-like planets are common in a low density environment" by Gratton et al.

Reviewer: Sean Raymond (you can identify me to the authors)

This paper focuses on the detection and detectability of gas giant planets at Jupiter- to Saturn-like orbital distances orbiting a carefully selected sample of stars in the Beta Pictoris moving group. The authors use various methods to claim the existence (or impossibility of existence) of 'Jupiter-like' planets and calculate a high occurrence rate of such planets in their sample.

Overall, I find the paper to be interesting and novel, with important implications for planet formation and detection. I think the paper is worthy of publication in Nature Communications once a few issues are resolved.

When reading the paper, two significant issues came to mind. The first relates to the three or four systems that are excluded from analysis because their 'Jupiter-like' zones are rendered unstable due to the dynamical influence of stellar or brown dwarf companions. The authors' calculations were performed assuming that all companions had an orbital eccentricity of 0.67. While 0.7 is indeed the median eccentricity expected for an 'isotropic' velocity distribution of bound objects, there could easily be one or more companions whose orbits are in fact much less eccentric. Given just a few objects, it is conceivable that all of them happen (by chance) to have near-circular orbits. Unless there are direct (measured) constraints on the eccentricity, I recommend including the possibility of circular orbits into the statistics, perhaps by widening the range of plausible giant planets occurrence rates. A second but related issue is simply small-number statistics. Given the very limited sample size of this study, what are the odds of this simply being a statistical fluke? A little more exploration into that possibility would make their results more robust. A final issue that I think needs addressing is the Figures, which could use more clarity (perhaps in the form of legends) to make them easier for the reader to interpret.

I also have a number of detailed comments below.

Detailed comments:

- The term "Jupiter-like" is quite vague and it would be helpful if it were defined sooner.

- The information/perspective in the referenced planetplanet blog post was included in two review papers, which could be cited instead: Raymond, Izidoro & Morbidelli (2020) and Raymond & Morbidelli (2022):

<https://ui.adsabs.harvard.edu/abs/2020plas.book..287R/abstract> and

<https://ui.adsabs.harvard.edu/abs/2022ASSL..466....3R/abstract>

- Table 1 contains fewer studies than I would have expected. How does it compare with the table compiled in Miret-Roig et al (2022)? For instance, I don't see references to Fernandes et al (2019) or the brand-new Lagrange et al (2023) paper.

- Top of page 2: "Models predict that giant planets should easily form around solar type stars through the core-accretion mechanism^{12, 13} and that the final semi-major axis distribution should be a consequence of the position of the ice line¹³, only partially modified by migration¹⁴". I understand that this paragraph is trying to create tension between models predicting many gas giants and observations finding fewer. However, it's worth keeping in mind that the disk mass plays a key role, since a massive core must form quickly enough to accrete gas from the disk. I would also make sure to cite Bitsch et al (2015) for the migration issue, although they found that large-scale migration was likely common.

- Same paragraph: to my knowledge, Suzuki et al (2016 and 2022) is the latest in microlensing planet statistics

- Introduction. It seems like the authors should discuss the role of stellar mass here, as they are focusing on higher-mass stars for which the giant planet occurrence rate is known to be significantly higher than for Solar-mass stars (e.g. Johnson et al 2010; see their Fig 4). I see that this is mentioned later but may be worth mentioning here. Metallicity is of course another key parameter that is worth mentioning, as I assume it is well measured for this sample.

- Figure 1. A legend on the figure would make it much easier to understand. For instance, what are the grey symbols? Are they just blue ones hiding behind the orange region?

- Page 3. The authors assumed an eccentricity of 0.67 for each stellar or BD companion, which must introduce a significant bias. Even if those companions have an isotropic velocity distribution such that the median expected eccentricity would be ~ 0.7 , a fraction of those systems have lower eccentricities and may allow for the stability of Jupiter-like planets. It seems worth including any stars for which the Jupiter-like region *could* be stable if the companion's orbit is circular. And, on a more general note, are there not strong enough constraints from Gaia and radial velocities to constrain the companions' orbits directly?

- Can you include a note in Fig 1 of which companions make the Jupiter-like zone unstable?

- Fig 2 needs to be explained much more clearly. Please use legends on the plot to make it easier to follow. Also, please redefine PMA in the caption. Am I understanding right that the PMA signal may come from planets along the blue lines but outside of the region of detection of the red/green lines?

- Can the authors expand on the idea that two Jupiter-mass planets might explain the case of HIP 10679?

Dear Reviewers,
we thank you very much for your comments that help in making the paper more straightforward and readable. In the following we give detailed answers to all points made. Modifications in the text are in boldface.

=====

REVIEWER COMMENTS

Reviewer #1 (Remarks to the Author):

Review of Nature Comms paper:
"Jupiter-like planets are common in a low density environment" by Gratton et al. (2023)

The authors have reviewed measurements of the occurrence rates of Jupiter-like planets from various literature sources, which together suggest that Jupiter-like planets are rare, occurring around less than 20% of stars. The authors have also considered the occurrence rates of Jupiter-like planets (mass > 0.1 M_{Jup} orbiting 3.0 to 12.0 au from their star) in a nearby stellar association: the Beta Pic Moving Group (BPMG), using existing studies and data from the Gaia spacecraft. Together, those data suggest that Jupiter-like planets in the BPMG are very common (occurrence rate almost 100%). Hence the authors conclude that Jupiter-like planets are common in low-density environments such as BPMG.

The overall methods of the paper are satisfactory, the results are statistically robust, and the conclusion appears to be supported. I do have comments that will perhaps affect the quoted occurrence rate but that will not change the conclusion.

The work is important because it contrasts with existing studies with only partially overlapping parameter spaces. There is also ample relevant discussion interpreting the new results and comparing them to the literature. It deserves to be published in Nature Comms.

Detailed comments follow.

1. The studies quoted in Table 1 have used different working definitions of Jupiter-like planets to the one used in this work. All studies use a narrower mass range (typically beginning at 1.0 M_{Jup}) and a narrower orbital radius than this work. Given the broader mass range in this work, it naturally follows that the occurrence rate found here will be higher. It is also known and expected that the occurrence rate of planets is a function of semi-major axis (or orbital radius); the definition in Cumming et al. is entirely non-overlapping with this work, making a direct comparison difficult, and for the other sources it naturally follows that the authors should find more companions in their broader semi-major axis range than are found in those literature studies. I would like to see a more thorough discussion and calculation of both of the above effects on the occurrence rate comparison.

A. We agree that this point should be elaborated further. While our text was a bit confused, in practice we adopted a definition of Jupiter-like planets as

objects with a mass in the range 1-10 M_{Jup} and with semimajor axis in the range 3-12 au. The lower limit in mass was adopted because small mass objects

beyond the ice line are very difficult to discover with current techniques. On the other hand, previous papers use different definitions; we spelled these in

Table 1. Cuts and extrapolations are required to obtain homogeneous values. In the case e.g. of Cumming et al. (2008), integration of the best power laws over

masses and semi-major axis indicates that 5.5% of the solar type stars should have a Jupiter-like planets according to our definition. This is however an

extrapolation, because the upper limit of the period range considered by these authors correspond to a semimajor axis of 3.1 au for a star of 1 Mo. There

is more overlap with the sample of Wittenmyer et al. (2016) that extends the semi-major axis range up to 7 au. Results by this last paper are broadly

consistent with those of Cumming, with a higher incidence rate by about 25% (well within the error bars) if the same range of masses/separation are considered. Zhu (2022) gives a higher incidence of giant planets than expected using the best value for the exponents in the Cumming et al.

power laws, by a factor of two if we consider the total population of planets and 1.6 if we rather consider the stars that have at least one planet. This

corresponds to the incidence of $6.9 \pm 1.8\%$ of planets with mass in the range 3-10 M_{Jup} and semi-major axis 3-10 au from Figure 4 of Zhu

paper. We modified the text of the Introduction to include this discussion. We can also notice that if we limit the mass range to 3-10 M_{Jup}, Zhu gives a

frequency of only 2.2+0.9-0.7% of the stars having companions in the range 3-10 au. A similar value (1.9%) has been obtained by Fernandes et al. (2019), a

reference that we added in this revised version. We This makes very unlikely (probability of 6.0×10^{-5}) the detection of 5 companions over a sample of 20

stars as found in the BPMG. A short text spelling this was introduced in the section "Results".

2. Regarding the Bryan et al. study, I agree with the authors that the conditional probability of a particular star hosting a cold Jupiter given the existence of other planets interior to that cold Jupiter's orbit is probably not random. However, I would like to see the evidence or a reference.

A. References are the same Bryan et al study, and in addition Knutson et al. (2014), Zhu et al. (2018), Bryan et al. (2019), Rosenthal et al. (2021), Zhu (2022). They are now given in the text.

3. Regarding the mass ratio distribution of companions to stars in the 1.0 to 2.5 M_{sun} mass range, the authors appear surprised that the distribution is skewed towards companions in the q=0.2 to 1.0 range. In fact, this distribution has been well measured and discussed (e.g. Moe & Di Stefano 2017 for a range of primary masses, Murphy et al. 2018 for ~1.5 to 2.0-M_{sun}

primaries). The latter study also shows the 'thermal distribution' of orbital eccentricities (attributed to Ambartsumian 1935, cited by the authors) to be a poor representation of the observations, hence the average eccentricity of 0.67 is a poor choice to use in the authors' simulations of dynamical stability. The 'average', taken as the eccentricity where the cumulative distribution reaches 0.5, is measured by Murphy et al to be closer to 0.33. Further, a power law $p(e) \sim e^{-\alpha}$ with $\alpha=0.1$ is a better fit than using a simple average, and a two-parameter Gaussian distribution with $\mu=0.36$ and $s=0.27$ would be better still.

A. For what concern the mass, what we wish to notice here is the scarcity of low-mass stellar companions. Only 8 out of the 23 stellar companions (35%) are M-type stars (here $M < 0.5 M_{\odot}$). For comparison, the fraction of objects with this mass is in the range 62-77% for the most commonly considered stellar initial mass functions (Chabrier2003, Chabrier2005, Kroupa2001). We respelled this sentence to clarify this. For what concerns

eccentricity, for one of the critical objects (PZ Tel) there is an orbit determination with an eccentricity of 0.52 (Franson &

Bowen 2023). For the remaining ones, we notice that the eccentricity distribution depends on the semimajor axis (see Hwang et al. 2022, in addition to Murphy

et al. 2018). The value of $\alpha=0.33$ obtained by Murphy et al. refers to rather compact binaries (period in the range 100-1500 d, semimajor axis < 3 au); for

separation around 100 au, a uniform distribution of eccentricities ($\alpha=0.5$) is a better representation, while for even larger separation (> 300 au) the

distribution is thermal or even suprathermal. For these targets, we then now assume an average value appropriate for the apparent separation of the targets

given by the relation:

$$= -0.0117 \sim \log(a)^3 + 0.0529 \sim \log(a)^2 + 0.07 \sim \log(a) + 0.3226$$

that was obtained combining the eccentricity distribution for close binaries obtained by Murphy et al. with that for wide binaries by Hwang et al. 2022.

In addition, strictly speaking, the relation between the average projected separation and the semi-major axis that we used to derive this last quantity

for several companions depends on the assumed eccentricity (Brandeker et al. 2006), so an iterative procedure should be required here. However, corrections

to the value of the eccentricity that are obtained by considering this fact are always < 0.005 , and the variation of the semi-major axis are always $< 5\%$

for our targets. We will neglect these second order effects. While these assumptions are somewhat different from those original done, there is no variation

about which of the target can have Jupiter-like planets on stable orbits. The text has been updated to take this into consideration.

4. For clarity and readability, I would suggest the following. After the sentence "The median probability that a planet in the range of separation 3-12 au around one of 16 stars possibly hosting Jupiter-like planets is not behind the coronagraph in at least one of the epochs when it was observed is 39%" the authors might want to add: "that is, the probability of detecting a planet at that separation around one of the 16 stars, given the sampling, is 39%." If that is not the intended meaning, then I fear the original wording is not very clear.

A. This is indeed the the intended meaning. We added the sentence proposed by the referee.

5. In figure 2, the solid "red" line actually appears orange.

A. Ok, we modified the caption accordingly.

6. In the final paragraph before the mass correction, the authors calculate the frequency of stars hosting Jupiter-like planets to be $44 \pm 18\%$, based on 7 out of 16 stars having them. The uncertainty here might not have been calculated correctly. The stars here represent Bernoulli trials: each draw represents a discrete probability distribution, which has a value of 1 ('star has planet') with probability p , and a value of 0 ('star has no planet') with probability $q=1-p$. The uncertainty is then $\sqrt{p \cdot q / n} = \sqrt{0.44 \cdot (1-0.44) / 16} = 0.124$. Hence the frequency is $(44 \pm 12)\%$.

A. Answering to comments made by another referee, we now extended the sample to stars more massive than 0.8 Mo. The statistics is now 7 out of 17, that using the approach described by the referee means a frequency of $(41 \pm 12)\%$

Following corrections to the above (the comparisons in point 1, the simulations in point 3, and the uncertainty in point 6), I anticipate that the conclusion will remain the same but the significance may differ. I look forward to seeing the update.

Reviewer #2 (Remarks to the Author):

The paper presents the fraction of solar-type stars hosting a "Jupiter-like" planet in the Beta Pictoris young moving group and concludes that the fraction is higher than found in the field.

My biggest concern is that this statistic is found by combining both confirmed *and undetected* Jupiter-like exoplanets within a small, biased sample of stars. In particular, that the number of candidate exoplanets doubles the sample of confirmed ones (4 confirmed plus 4 candidates), artificially enhancing the principal statistic supporting the claim, and ultimately that the claim is based on small number statistics.

A. This is not exactly true. We use two different approaches: (i) using the detected companions + corrections due to visibility in orbital phase; (ii) using

the detected companions + undetected ones inferred from PMA (PMA should be applicable to the objects that are not detectable because the planet is in the

wrong phase). The two methods yield frequencies of $36 \pm 9\%$ and $41 \pm 12\%$, that agree well within the errors. To better clarify this point, we restructured

the discussion about the orbital sampling effect in a single subsection subdivided into a brief introduction, a detailed description of the two approaches

used, and a brief concluding section. For what concerns the size of the sample, it is true that the statistics is not huge but it is also neither null

and allows outlining the apparent inconsistency that is already emerging from available data. This is now discussed in an individual subsection called:

"Statistical relevance of the observed frequency of Jupiter-like planets in our sample" where we describe our Monte Carlo approach to this issue.

Major comments:

- The term "low density environment" should be clearly defined early in the paper, as it could be interpreted in myriad ways depending on the subfield of expertise of the reader.

A. We agree. We define here {\it low-density} environment a star forming region with less than 300 M \odot . This is now specified in the Discussion section.

- Figure 1 is a bit cryptic. It is not clear what is the message that it is trying to convey with the orange rectangle. It would be more digestible if the rest of the exoplanet population was shown to provide context. At minimum, a color legend would be immensely helpful.

A. We revised this figure including a colour legend. The orange rectangle corresponds to the definition of Jupiter-like planets, as spelled in the last

sentence of the caption. We prefer not to draw the whole exoplanet population because it is not obtained from complete samples, so their distribution may be misleading.

- Assuming a point estimate of $e=0.67$ for the eccentricity of Jupiter-like planets is not well motivated and unlikely to be robust. A better reference would be Bowler et al. 2020 from which the authors could draw a distribution of young, directly-imaged planets, for a close observational match to the parameters of their sample. Alternatively, a Monte Carlo simulation would serve a similar purpose to achieve robustness.

- The assumption of eccentricities for Jupiter-like planets is inconsistent across the paper: a circular orbit assumption is used to estimate the fraction of time a planet spends behind a coronagraph, and is therefore undetectable, yet a fixed eccentricity of $e=0.67$ was assumed to calculate orbital stability.

A. We answer these two points together. We are not assuming an eccentricity of $e=0.67$ for the Jupiter-like planets. Rather, we were using this value for

the stellar/BD companions when estimating the range of semi-major axes where a planet may reside. We however revised this point answering to a point made

by another referee. Repeating the argument, for one of the critical objects (PZ Tel) there is an orbit determination with an eccentricity of 0.52 (Franson &

Bowen 2023). For the remaining ones, we notice that the eccentricity distribution depends on the semimajor axis (see Hwang et al. 2022, in addition to Murphy

et al. 2018). The value of $e=0.33$ obtained by Murphy et al. refers to rather compact binaries (period in the range 100-1500 d, semimajor axis <3 au); for

separation around 100 au, a uniform distribution of eccentricities ($e=0.5$) is a better representation, while for even larger separation (>300 au) the

distribution is thermal or even suprathermal. For these targets, we then now assume an average value appropriate for the apparent separation of the targets

given by the relation:

$$= -0.0117 \sim \log(a)^3 + 0.0529 \sim \log(a)^2 + 0.07 \sim \log(a) + 0.3226$$

that was obtained combining the eccentricity distribution for close binaries obtained by Murphy et al. with that for wide binaries by Hwang et al. 2022.

In addition, strictly speaking, the relation between the average projected separation and the semi-major axis that we used to derive this last quantity

for several companions depends on the assumed eccentricity (Brandeker et al. 2006), so an iterative procedure should be required here. However, corrections

to the value of the eccentricity that are obtained by considering this fact are always <0.005 , and the variations of the semi-major axis are always $<5\%$

for our targets. We will neglect these second order effects. While these assumptions are somewhat different from those originally done, there is no variation

about which of the targets can have Jupiter-like planets on stable orbits. The relevant text has been updated to take this into consideration

Rather, we considered the eccentricity of planets when estimating the fraction of time spent behind the coronagraphic mask, in order to correct for what we

called "orbit sampling effect". In the first version of the paper we assumed circular orbits - that is zero eccentricity - for the planets. Of course, we

do not expect the orbits to be really circular. To answer this point made by the referee we wrote a brief review of what is known about the eccentricities

of Jupiter-like planets in the Methods section. We found that this can be represented by a uniform distribution over the range 0-0.5.

With this in mind, we repeated our estimate of the probability of observing a companion far enough from the star to be detected in the HCI observation.

We considered both the case of circular orbits and of eccentric orbits with eccentricities uniformly distributed between 0 and 0.5. The two results are

quite similar; we however used the results obtained with eccentric orbit when examining the "orbit sampling effect". The relevant text has been re-written to explain all this.

- Since the claim of high Jupiter-like frequency around solar-type stars relies on the detectability fraction estimated through their simulation, a much more detailed description and discussion of said simulation is warranted, as well as clear parameter boundaries for its applicability (e.g., range of primary and companion masses where the simulation and

observations are sensitive).

- Does the "95% level of confidence" refer to a confidence interval? If yes, how was this defined? Is it a percentile of the bootstrapped samples? Do 95% of the Monte Carlo runs yield a frequency of Jupiter-like planets between 58 and 100%?

A. We answer these two points together. We notice that all Jupiter-like planets detected by HCI are more massive than 4 MJ and were at a separation >150 mas

at the epoch of observation. We then consider two corrections. The first one is related to the fraction of time spent by planets with mass >4 MJ at a separation >150 mas. This is computed considering the probability that a HCI observation of the target was acquired when the planet was at >150 mas from the

star and taking the median value over the sample. This value is compared with "indirect detections" that are obtained considering the objects with $\text{SNR}(\text{PMa}) > 3$,

that also corresponds to planets with masses >4 MJ for the current sample. The two values agree well and indicate that $39 \pm 12\%$ of the stars in the sample

have a Jupiter-like planet with a mass >4 MJ. The second correction takes into account of the Jupiter-like planets with mass in the range 1-4 MJ. We found

that for reasonable mass distributions of the Jupiter-like planets, $37 \pm 9\%$ of the Jupiter-like planets have a mass >4 M_J . We then run a Monte Carlo

simulation that estimates the probability of detecting the observed fraction of planets in our sample. We rewrote the description of the procedure used to

derive the best value and the 95% confidence level for the frequency of Jupiter-like planets around stars in the BPMG sample and deserved to it an individual subsection called: "Statistical relevance of the observed frequency of Jupiter-like planets in our sample". We hope that the new description is

clear enough to be repeatable and answer the points made by the referee.

- The direct comparison between direct imaging and RV samples in the Discussion needs to be taken with a grain of salt. The BPMG is a young association, whereas the RV-detected giant planet sample is primarily composed of field-age stars since young stars have high levels of RV jitter and have been selected out of RV surveys for decades. This is an important bias to take into account (in addition to the difference in median primary mass as noted by the authors) since dynamical evolution can dramatically affect the orbital parameters of a given system and it is more likely that field-age systems have undergone chaotic dynamical evolution than systems in a young moving group.

A. Sure. This is exactly the second point made in the discussion

- The discussion involving the "conundrum with the low frequency found by radial velocity surveys" needs significant development.

A. This point is generic, not easy to reply. We considered four possible explanations (mass dependence, dynamical evolution, different bias against outer

stellar/BD companions, environment effects on planet formation), as well as a combination of them. The purpose of this paper is to raise the possibility

that environment plays an important role determining the frequency of Jupiter-like planets, not to find the correct answer to this issue, that requires

extensive exploration of dynamical models well beyond what can be done in a short paper such as the present one.

Minor comments:

- Please reference Franson et al. 2023 as well for the discovery of AF Lep b.

A. Done. Actually, the Franson et al. paper was submitted one month after the discovery papers by Mesa+ and De Rosa+, and as today it is not yet published

on a refereed journal

- The down selection of BPMG sample from 164 to the final 14 does not justify why they only pick stars more massive than the Sun if their goal is to explore "Sun-like" stars, especially since their analysis suffers from small number statistics.

A. In order to make the sample more appropriate for "Sun-like" stars we changed the lower mass limit from 1.0 to 0.8 M_\odot . The changes are small because

most of the stars with mass in the range 0.8-1.0 M_\odot in the BPMG have a stellar companion near the region appropriate for Jupiter-like planets, making

it impossible to have them. There are then only four additional stars around which we might find Jupiter-like planets, bringing the total number to 20.

Only two of these four stars are in Hipparcos, so the others lacks an estimate of the PMa; we however updated the paper with this larger sample.

- Figure 1 brings into question what the authors may mean with the word "companion". This word is typically used for stellar or brown dwarf companions. If used for "planetary companion," especially since they use the term to refer to objects within the orange rectangle, it would be helpful for the reader that the authors specify this explicitly.

A. OK; we now specify explicitly that this plot includes both stellar and substellar companions.

- Says: "We considered companions objects with full (5-elements) astrometric solution within 10 arcmin from each stars, whose relative projected velocity is below the escape velocity". Should say "of the cluster" at the end of this sentence.

A. The BPMG is not bound, so there is no meaning to refer to escape velocity from the cluster. What we consider here is escape velocity from the star at

the projected distance of the object. This is now spelled explicitly.

- This statement: "Objects with RUWE > 1.4 are likely binaries" needs a reference. I believe Torres & Stassun 2018 should do.

A. We already cited Belokurov+ 2020, that is the major reference paper about RUWE and binarity. We do not find any Torres & Stassun 2018 paper on

ADS,

but likely the referee meant Stassun & Torres (2021, ApJ, 907, L33), that is a very appropriate reference here too. We now included citation of that paper.

- The paper would be strengthened with Figures exploring the sample in more detail, for example showing the distributions of mass and distance.

A. We added (in the Methods Section) a figure showing the distribution of the BPMG stars with $M > 0.8 M_{\odot}$ in the distance mass plane. We used different symbols

for stars around which a Jupiter-like planet has been detected using HCI, there is indication of a similar companion from PMA, there is a binary companion

that makes unstable the orbits of potential Jupiter-like planets, there is no detection and those lacking PMA data. Jupiter-like planets have been discovered

around the three closest stars of the BPMG with mass above solar. Those stars where there is indication for similar (yet undetected) companions from PMA are

intermediate in distance and mass. These facts suggest that the lack of detections in the remaining stars may be due to selection effects.

- Please justify the need for Figure 1 (why is mass ratio vs semi-major axis an interesting space to explore?).

A. This is the fundamental plane to understand the formation of companions; to show this we added citation of Mordasini et al. (2009, A&A 501, A3) about planet formation.

- "The stellar companions are confined in a quite narrow range of mass ratios ($0.2 < q < 1.0$)" -- Since mass ratio spans from 0-1, $q = 0.2-1.0$ is most of the mass ratio range.

A. What we wish to notice here is the scarcity of low-mass stellar companions. Only 8 out of the 23 stellar companions ($\sim 35\%$) are M-type stars (here $M < 0.5 M_{\odot}$). For comparison, the fraction of objects with this mass is in the range 62-77% for the most commonly considered stellar initial mass functions (Chabrier2003, Chabrier2005, Kroupa2001). We respelled this sentence to clarify this.

- Multiple typos were found throughout the text.

A. We further revised the text looking for typos.

=====

Reviewer #3 (Remarks to the Author):

Review of "Jupiter-like planets are common in a low density environment" by Gratton et al.

Reviewer: Sean Raymond (you can identify me to the authors)

This paper focuses on the detection and detectability of gas giant planets at Jupiter- to Saturn-like orbital distances orbiting a carefully selected sample of stars in the Beta Pictoris moving group. The authors use various methods to claim the existence (or impossibility of existence) of 'Jupiter-like' planets and calculate a high occurrence rate of such planets in their sample.

Overall, I find the paper to be interesting and novel, with important implications for planet formation and detection. I think the paper is worthy of publication in Nature Communications once a few issues are resolved.

When reading the paper, two significant issues came to mind. The first relates to the three or four systems that are excluded from analysis because their 'Jupiter-like' zones are rendered unstable due to the dynamical influence of stellar or brown dwarf companions. The authors' calculations were performed assuming that all companions had an orbital eccentricity of 0.67. While 0.7 is indeed the median eccentricity expected for an 'isotropic' velocity distribution of bound objects, there could easily be one or more companions whose orbits are in fact much less eccentric. Given just a few objects, it is conceivable that all of them happen (by chance) to have near-circular orbits. Unless there are direct (measured) constraints on the eccentricity, I recommend including the possibility of circular orbits into the statistics, perhaps by widening the range of plausible giant planets occurrence rates.

A. This is a point also made by the two other referees and clearly needed consideration. So we decided to use more appropriate eccentricity values for

each star in the sample. For one of the critical objects (PZ Tel) there is an orbit determination with an eccentricity of 0.52 (Franson & Bowen 2023). For

the remaining ones, we notice that the eccentricity distribution depends on the semimajor axis (Murphy et al. 2018, Hwang et al. 2022). The value of $e = 0.33$

obtained by Murphy et al. refers to rather compact binaries (period in the range 100-1500 d, semimajor axis < 3 au); for separation around 100 au, a uniform

distribution of eccentricities ($e = 0.5$) is a better representation, while for even larger separation (> 300 au) the distribution is thermal or even suprathermal. For these targets, we then now assume an average value appropriate for the apparent separation of the targets given by the relation:

$$e = -0.0117 \sim \log(a)^3 + 0.0529 \sim \log(a)^2 + 0.07 \sim \log(a) + 0.3226$$

that was obtained combining the eccentricity distribution for close binaries obtained by Murphy et al. with that for wide binaries by Hwang et al. 2022.

In addition, strictly speaking, the relation between the average projected separation and the semi-major axis that we used to derive this last quantity

for several companions depends on the assumed eccentricity (Brandeker et al. 2006), so an iterative procedure should be required here. However, corrections

to the value of the eccentricity that are obtained by considering this

fact are always $<0.005\%$, and the variation of the semi-major axis are always $<5\%$

for our targets. We will neglect these second order effects. While these assumptions are somewhat different from those original done, there is no variation

about which of the target can have Jupiter-like planets on stable orbits. The text has been updated to take this into consideration.

A second but related issue is simply small-number statistics. Given the very limited sample size of this study, what are the odds of this simply being a statistical fluke? A little more exploration into that possibility would make their results more robust.

A. These odds are taken into account by the Monte Carlo approach we adopted to establish the statistical meaning of the result. Since the assumption

made concerning this method were not well understood, we spelled there more clearly in a new individual subsection called: "Statistical relevance of the

observed frequency of Jupiter-like planets in our sample"

A final issue that I think needs addressing is the Figures, which could use more clarity (perhaps in the form of legends) to make them easier for the reader to interpret.

A. We added legends to the figures.

I also have a number of detailed comments below.

Detailed comments:

- The term "Jupiter-like" is quite vague and it would be helpful if it were defined sooner.

A. While our text was a bit confused, in practice we adopted a definition of Jupiter-like planets as objects with a mass in the range $1-13 M_{\text{Jupiter}}$ and

with semimajor axis in the range $3-12$ au. The lower limit in mass was adopted because small mass objects beyond the ice line are very difficult to discover

with current techniques. This is now spelled clearly. Since each author used a different definition, we commented about results that are obtained with this

definition in the different studies in the Introduction.

- The information/perspective in the referenced planetplanet blog post was included in two review papers, which could be cited instead: Raymond, Izidoro & Morbidelli (2020) and Raymond & Morbidelli (2022): <https://ui.adsabs.harvard.edu/abs/2020plas.book..287R/abstract> and <https://ui.adsabs.harvard.edu/abs/2022ASSL..466...3R/abstract>

A. We thank the referee for indicating these references. We added them in the Introduction

- Table 1 contains fewer studies than I would have expected. How does it compare with the table compiled in Miret-Roig et al (2022)? For instance, I don't see references to Fernandes et al (2019) or the brand-new Lagrange et al (2023) paper.

A. We thank the referee for indicating these references. The frequency of Jupiter-like planets obtained by Fernandes et al.(2019) is low; integrating over our definition of Jupiter-like planets, it is only 4.2%. Lagrange et al. paper is not yet accepted and does not give enough detail for the present

purposes; however, we cite it now as it warns against the assumption that the mass distribution is independent of period, as rather assumed by Cumming et al. (2008) and Fernandes et al. (2019)

- Top of page 2: "Models predict that giant planets should easily form around solar type stars through the core-accretion mechanism^{12, 13} and that the final semi-major axis distribution should be a consequence of the position of the ice line¹³, only partially modified by migration¹⁴". I understand that this paragraph is trying to create tension between models predicting many gas giants and observations finding fewer. However, it's worth keeping in mind that the disk mass plays a key role, since a massive core must form quickly enough to accrete gas from the disk. I would also make sure to cite Bitsch et al (2015) for the migration issue, although they found that large-scale migration was likely common.

A. We did not wish to create tension between theory and observation; we only record existing results. Following the referee advice, we included citation of

Bitsch et al (2015).

- Same paragraph: to my knowledge, Suzuki et al (2016 and 2022) is the latest in microlensing planet statistics

A. Yes, we knew. But the Suzuki statistics is for a typical microlens, that is much less massive than a Solar type star. If we integrate their data,

the typical frequency of Jupiter-like planets is only 0.7%. This low value is likely due to the low relative frequency of Jupiter-like planets around

M-stars. We decided not to include Suzuki et al. papers here because they seem inappropriate to discuss solar-type stars.

- Introduction. It seems like the authors should discuss the role of stellar mass here, as they are focusing on higher-mass stars for which the giant planet occurrence rate is known to be significantly higher than for Solar-mass stars (e.g. Johnson et al 2010; see their Fig 4). I see that this is mentioned later but may be worth mentioning here.

A. Yes. We added a sentence recalling these dependences in the Introduction. The mass dependence was already considered in the Discussion section; it

might contribute to explain the apparent discrepancy in the frequency of Jupiter like planets in the BPMG and from RVs, but the effect is likely not enough

to completely explain it. By the way, in order to answer to a point raised by another referee, we extended the mass range to include all BPMG stars with

mass $M > 0.8 M_{\odot}$. While there are small changes in the numbers, these do not modify the main results of the paper.

- Metallicity is of course another key parameter that is worth mentioning, as I assume it is well measured for this sample.

A. We introduced a paragraph in the Methods section discussing the metallicity of the BPMG. Since it is roughly solar, it cannot be an explanation of the discrepancy found. We added a few words to say this in the Discussion section.

- Figure 1. A legend on the figure would make it much easier to understand. For instance, what are the grey symbols? Are they just blue ones hiding behind the orange region?

A. We added a legend to this figure.

- Page 3. The authors assumed an eccentricity of 0.67 for each stellar or BD companion, which must introduce a significant bias. Even if those companions have an isotropic velocity distribution such that the median expected eccentricity would be ~ 0.7 , a fraction of those systems have lower eccentricities and may allow for the stability of Jupiter-like planets. It seems worth including any stars for which the Jupiter-like region *could* be stable if the companion's orbit is circular. And, on a more general note, are there not strong enough constraints from Gaia and radial velocities to constrain the companions' orbits directly?

A. This is discussed above.

- Can you include a note in Fig 1 of which companions make the Jupiter-like zone unstable?

A. We mark with a red circle those companions that make the Jupiter-like zone unstable

- Fig 2 needs to be explained much more clearly. Please use legends on the plot to make it easier to follow. Also, please redefine P_{Ma} in the caption. Am I understanding right that the P_{Ma} signal may come from planets along the blue lines but outside of the region of detection of the red/green lines?

A. We added labels explaining the meaning of each line in every panel. We redefined the P_{Ma} in the caption. The object responsible for the P_{Ma} should be

close to the blue line and below the the red/green lines

- Can the authors expand on the idea that two Jupiter-mass planets might explain the case of HIP 10679?

A. Very simply, the observed P_{Ma} for a star having several companions is the vectorial sum of the P_{Ma} due to individual companions. The signal due to two

planets, each one with a mass smaller than that indicated by the blue line, may combine to produce a signal similar to an individual planet with a mass as

indicated by the blue line, if the individual P_{Ma} vectors have a similar direction. Note that a combination of the signal of two planets may also lead to a

cancellation, that is a P_{Ma} lower than expected for the individual objects, if the individual P_{Ma} vectors are directed at very different angles one from the

other. We added a note to explain this in the text.

REVIEWERS' COMMENTS

Reviewer #1 (Remarks to the Author):

The authors have provided reasonable replies and actions to my initial comments. I have further comments. The article deserves to be published.

Reviewer #2 (Remarks to the Author):

Thank you to the authors for taking into account my comments and revising their draft. I appreciate the new section on orbital

sampling effects and the extended discussion on the orbital effects from the choice of eccentricity. I only have a couple more minor comments on the text.

When describing the Bryan et al. 2015 study:

"A higher total occurrence rate ... Jupiter, is in fact likely not random"

These sentences are implying that the Bryan study was biased where in reality they were looking for precisely the effect they found, so it's unclear what point the authors are trying to make.

"The final sample is made of 30 stars; membership of some of these stars to the BPMG is uncertain."

Consider rephrasing this sentence. In the Methods section, the initial sample of 38 stars gets trimmed down to 30 partially because of the membership. The sentence as is weakens the conclusion because it suggests that a fraction of the 30-star sample is potentially not part of BPMG.

"The fraction of single stars ($37 \pm 12\%$) is lower than that found for solar-type stars by Raghavan et al."

Add here that this is a field study, so the higher binary fraction the authors are finding is consistent with Raghavan et al. since we expect some fraction of binaries to dissipate as they age.

Reviewer #3 (Remarks to the Author):

I appreciate the changes made by the authors in response to my comments and also to the comments of the other two referees. I'm happy with the modifications and think the paper can be published as is.

Reviewer #1 (Remarks to the Author):

The authors have provided reasonable replies and actions to my initial comments. I have further comments. The article deserves to be published.

A. Nothing to be done here

Reviewer #2 (Remarks to the Author):

Thank you to the authors for taking into account my comments and revising their draft. I appreciate the new section on orbital sampling effects and the extended discussion on the orbital effects from the choice of eccentricity. I only have a couple more minor comments on the text.

When describing the Bryan et al. 2015 study:

"A higher total occurrence rate ... Jupiter, is in fact likely not random"

These sentences are implying that the Bryan study was biased where in reality they were looking for precisely the effect they found, so it's unclear what point the authors are trying to make.

A. We rephrased this sentence to make it more clear as: "... However, while very interesting, this higher incidence may be not naively considered as an estimate of the overall frequency of Jupiter-like companions. ..."

"The final sample is made of 30 stars; membership of some of these stars to the BPMG is uncertain."

Consider rephrasing this sentence. In the Methods section, the initial sample of 38 stars gets trimmed down to 30 partially because of the membership. The sentence as is weakens the conclusion because it suggests that a fraction of the 30-star sample is potentially not part of BPMG.

A. We rephrased this sentence. "**The final sample is made of 30 stars. It can be noticed that membership of some of these stars to the BPMG is uncertain. We adopted a conservative approach, where some uncertain members around which no companion was detected were nonetheless kept in the final list; while this reduces the derived frequency of Jupiter-like planets in the BPMG, as we will see this has little impact on our main conclusion**."

"The fraction of single stars (37±12%) is lower than that found for solar-type stars by Raghavan et al."

Add here that this is a field study, so the higher binary fraction the authors are finding is consistent with Raghavan et al. since we expect some fraction of binaries to dissipate as they age.

A. We agree. We modified the sentence as follows: "...as expected because this last study refers to much older stars and we expect some fraction of binaries to dissipate as they age. It is similar..."

Reviewer #3 (Remarks to the Author):

I appreciate the changes made by the authors in response to my comments and also to the comments of the other two referees. I'm happy with the modifications and

think the paper can be published as is.

A. Nothing to be done here